# Dendro-dendritic cholinergic excitation controls dendritic spike initiation in retinal ganglion cells

A. Brombas[1,*], S. Kalita-de Croft[1,*], E.J. Cooper-Williams[1] & S.R. Williams[1]

The retina processes visual images to compute features such as the direction of image motion. Starburst amacrine cells (SACs), axonless feed-forward interneurons, are essential components of the retinal direction-selective circuitry. Recent work has highlighted that SAC-mediated dendro-dendritic inhibition controls the action potential output of direction-selective ganglion cells (DSGCs) by vetoing dendritic spike initiation. However, SACs co-release GABA and the excitatory neurotransmitter acetylcholine at dendritic sites. Here we use direct dendritic recordings to show that preferred direction light stimuli evoke SAC-mediated acetylcholine release, which powerfully controls the stimulus sensitivity, receptive field size and action potential output of ON-DSGCs by acting as an excitatory drive for the initiation of dendritic spikes. Consistent with this, paired recordings reveal that the activation of single ON-SACs drove dendritic spike generation, because of predominate cholinergic excitation received on the preferred side of ON-DSGCs. Thus, dendro-dendritic release of neurotransmitters from SACs bi-directionally gate dendritic spike initiation to control the directionally selective action potential output of retinal ganglion cells.

[1] Queensland Brain Institute, The University of Queensland, Brisbane, Queensland QLD 4072, Australia. * These authors contributed equally to this work. Correspondence and requests for materials should be addressed to S.R.W. (email: srw@uq.edu.au).

The algorithms underlying neuronal circuit-based computations are embedded in network connectivity, and implemented by the functional operations of component synapses and neurons. The retina provides an ideal neuronal circuit to investigate such algorithms[1], as the visual world is computed by a relatively simple three-layered network to drive action potential output in classes of retinal ganglion cells which signal salient visual features, such as the direction of image motion to the higher brain[2–9]. In the retina the direction of image motion is signalled by classes of direction-selective retinal ganglion cells (DSGCs), which generate robust patterns of action potential output in response to light stimuli moving across or within their receptive fields in a preferred direction, but respond weakly to identical stimuli moving in the opposite null direction[2–4,6,10].

The direction-selective action potential output of DSGCs is believed to be computed by the integration of a directionally un-tuned excitatory input, predominately mediated by glutamate release from bipolar cells[11–13], with a directionally tuned inhibitory synaptic input generated by the dendritic release of GABA from axonless feed-forward interneurons, termed starburst amacrine cells (SACs)[9,14–17]. SACs represent essential components of the direction-selective circuitry of the retina[18]. Both ON- and OFF-SACs are distributed, in a mosaic throughout the retina[19], and possess a unique radial dendritic morphology, in which each dendrite operates in electrical isolation[14,20,21]. Functionally, two-photon calcium imaging has revealed that SAC dendritic calcium responses are preferentially generated when light stimuli move from the soma towards terminal dendritic sites[14], a direction-selective calcium signal that is thought to generate the dendritic release of neurotransmitters from terminal dendritic synaptic output zones[22–24]. The directional tuning of DSGCs is disrupted by the pharmacological antagonism of GABA$_A$ receptors[23,25], evincing a prominent role of SAC-mediated synaptic inhibition. Consistent with this, electrophysiological and high-resolution morphological studies have demonstrated that the SAC-mediated inhibitory synaptic control of null direction light responses is mediated by a greater GABA$_A$ receptor-mediated synaptic conductance, synapse number, and distribution of dendro-dendritic synapses on the null side of DSGCs[16,17,26–28]. SACs are, however, not simply feed-forward inhibitory interneurons, as ultrastructural and functional evidence indicates that both GABA and acetylcholine (ACh) are co-released by SACs to drive postsynaptic inhibition and excitation[16,23,25–27,29–36]. The physiological role of the co-released neurotransmitter ACh is however less well understood, and controversy remains on the contribution of this feed-forward excitatory signal to the generation of light-evoked DSGC action potential output, and its role in the computation of direction selectivity[12,16,23,26,29,30,36–41]. Notably, SAC-mediated cholinergic signalling has been demonstrated to powerfully control the action potential output of DSGCs in response to time-varying visual stimuli, and act as an essential excitatory signal under mesopic, low contrast, conditions[40,41]. Light-evoked feed-forward SAC-mediated cholinergic excitation may therefore function to provide local dendritic depolarization that acts to gate bipolar-cell-mediated glutamatergic excitation of DSGCs[41], which is largely mediated by NMDA receptors[42]. Recent work has, however, highlighted that dendritic spike generation plays a major role in computation of direction selectivity[43–45]. Direct dendritic recordings have revealed that light-evoked action potential firing of rabbit DSGCs is driven by the initiation of a cascade of dendritic spike generators, a dendritic computation that is silenced by SAC-mediated inhibition[45]. As GABA and ACh are released from dendritic sites of SACs[16,23,25–27,33–36], these observations suggest that SAC-mediated cholinergic excitation may also function to control dendritic electrogenesis.

Here we use multi-site electrophysiological recording techniques to demonstrate that the dendro-dendritic release of ACh from SACs powerfully positively gates the initiation of dendritic spikes to control the stimulus-sensitivity, receptive field structure and the magnitude of light-evoked action potential output of DSGCs.

## Results

**Paired recordings from ON-SACs and ON-DSGCs.** The role of SACs in the directional selective circuitry of the retina has largely been explored with reference to ON-OFF-DSGCs, a cell-type which receives synaptic input in both the ON- and OFF-sublamina of the inner plexiform layer (IPL)[4–6]. In contrast, ON-DSGCs are exclusively driven by the ON pathway, and possess a dendritic tree restricted to the ON- sublamina of the IPL[4–6], providing a simplified circuit for the dissection of the role of SACs in the computation of direction selectivity. Direct electrical recording from the dendrites of ON-DSGCs has revealed that active dendritic integration plays an instrumental role in the generation and control of light-evoked action potential firing and the computation of direction selectivity[45]. To investigate the role that SAC-mediated cholinergic signalling plays in this computation we first made paired recordings from morphologically identified ON-SACs and ON-DSGCs maintained in fragments of the adult rabbit retina ex vivo (Fig. 1).

Paired recordings revealed that the generation of threshold electrogenesis in ON-SACs evoked either net excitatory or net inhibitory postsynaptic potentials (PSPs) in synaptically connected ON-DSGCs (Fig. 1; SAC current input: amplitude $= 0.65 \pm 0.06$ nA; duration $= 10$ ms; $n = 16$ pairs). In connections that exhibited excitatory PSPs, the selective pharmacological blockade of nicotinic ACh receptors (nAChRs) abolished the excitatory response to reveal postsynaptic inhibition (Fig. 1a-c; amplitude distribution in mecamylamine (MMA, 10 μM) significantly different from control, $P < 0.0001$; Kolmogorov–Smirnov test; median significantly different from control, $P < 0.0001$, Mann–Whitney test; $n = 9$). Conversely, the specific GABA$_A$ receptor antagonist GABAzine (10 μM) unmasked nAChR-mediated excitation in connections that exhibited net inhibition under control conditions (Fig. 1d,e; $P < 0.0001$; Kolmogorov–Smirnov test; median significantly different from control, $P < 0.0001$, Mann–Whitney test; $n = 6$). Pharmacologically isolated cholinergic excitatory and GABAergic inhibitory PSPs exhibited a characteristic difference in their time to onset following SAC activation, with the onset of GABAergic inhibitory PSPs leading that of cholinergic excitatory PSPs (Fig. 1f–h, Supplementary Fig. 1a–b; IPSP onset time $= 6.6 \pm 0.9$ ms; EPSP onset time $= 10.3 \pm 1.3$ ms, $P = 0.0432$; T $= 2.295$; onset time measured at 5% of PSP amplitude; IPSP rise time $= 8.9 \pm 1.9$ ms; EPSP rise time $= 5.8 \pm 0.5$ ms; rise time measured between 10 and 90% of PSP amplitude; $n = 14$). This difference could not be accounted for by a presynaptic mechanism, such as temporal jitter in the engagement of dendritic transmitter release, as the distribution of onset latencies around the mean was similar for pharmacologically isolated excitatory and inhibitory PSPs (Supplementary Fig. 1c). The onset time of SAC-evoked PSPs could however be influenced by the time course of transmitter diffusion from SAC dendritic sites. Previous studies have suggested cholinergic signalling may occur in a paracrine fashion in the retina, and so may not be reliant on structurally determined dendro-dendritic synapses[17,41], an idea consistent with the observed slower onset time of the pharmacologically

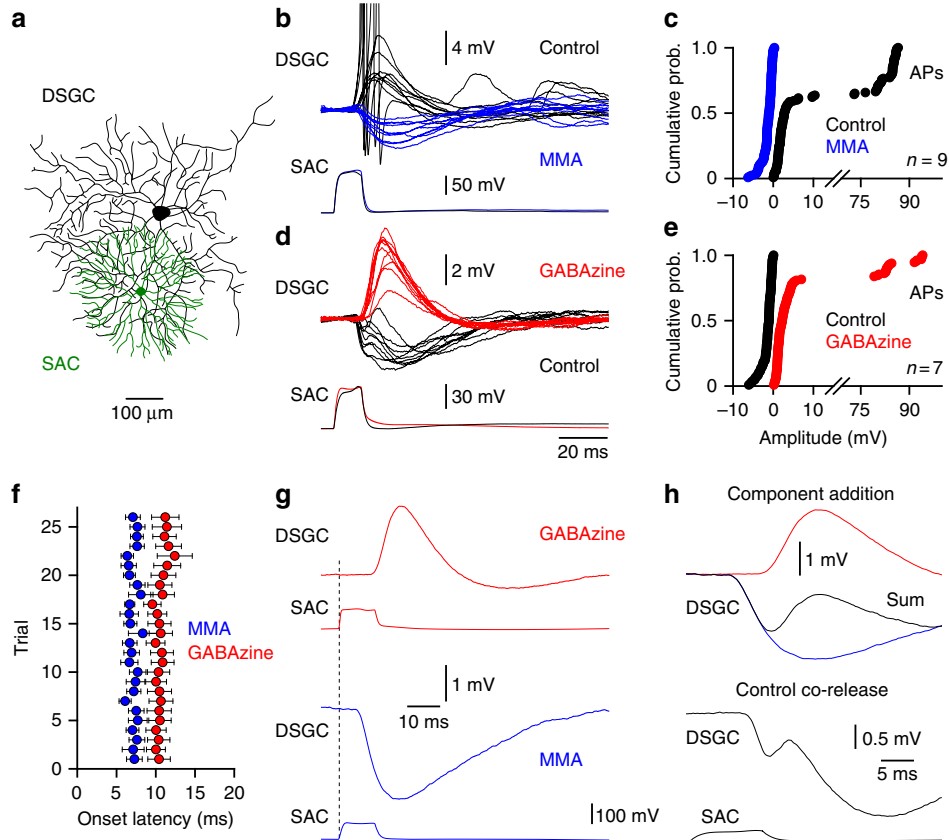

**Figure 1 | Starburst amacrine cell-mediated cholinergic excitation and GABAergic inhibition.** (**a**) Reconstruction of a connected pair of ON-SAC (green) and ON-DSGCs (black). (**b**) Paired recording revealed the obligatory co-release of ACh and GABA from ON-SACs. Overlain voltage traces show the transformation of SAC-driven synaptic excitation of an ON-DSGC to synaptic inhibition when nAChRs were blocked with Mecamylamine (MMA, 10 μM, blue traces; action potentials (APs) have been truncated for clarity). The lower traces show the somatic current-evoked excitation of the ON-SAC. (**c**) Cumulative probability distributions of the amplitude of each SAC-evoked PSP recorded under the indicated conditions, from the indicated number of paired recordings. (**d**) In a different pair of ON-SAC and ON-DSGCs net inhibition was generated under control conditions, which was transformed to excitation by the blockade of GABA$_A$ receptors (SR-95531 (GABAzine), 10 μM, red; morphologies shown in **a**). (**e**) Cumulative probability distributions of PSP amplitude under the indicated conditions. (**f**) Onset latency of pharmacologically isolated excitatory and inhibitory PSPs, recorded in GABAzine ($n = 7$) and MMA ($n = 7$), respectively. Onset latency was measured from the onset of the presynaptic driving current to 5% of the peak of the postsynaptic response. (**g**) Averaged pharmacologically isolated excitatory and inhibitory PSPs, the vertical dashed line indicates the onset time of the presynaptic driving current. (**h**) Upper overlain traces show the biphasic waveform (sum, black trace) produced by the arithmetic summation of isolated excitatory and inhibitory PSPs. The lower trace shows an averaged PSP recorded under control conditions, note the biphasic waveform.

isolated cholinergic PSPs observed here. We note, however, that previous paired recordings have revealed similar onset times of SAC-mediated cholinergic and GABAergic postsynaptic currents in DSGCs, when temporally overlapping excitatory and inhibitory synaptic currents were separated under somatic voltage-clamp by reversal potential[16]. Functionally, the different onset of dendro-dendritic excitation and inhibition observed here would be predicted to lead to the generation of biphasic compound PSPs, if both GABA and ACh are co-released from a single ON-SAC. Indeed when pharmacologically isolated excitatory and inhibitory components were arithmetically summed, a clear biphasic compound PSP was generated (Fig. 1h). Consistent with this, SAC-evoked PSPs recorded under control conditions frequently exhibited a biphasic waveform, that was clearly evident in single trials and when consecutive responses were digitally averaged (Fig. 1d,h, respectively). Together, these data reveal that ACh and GABA are co-released from single ON-SACs.

We next enquired if the synaptic impact of ON-SACs aligned with the directional tuning of postsynaptic ON-DSGCs (Fig. 2). To do this we mapped the action potential output of ON-DSGCs generated by light bars moved across their receptive fields and

made paired recordings from ON-SACs positioned close to the edge of the dendritic arbour first activated by light stimuli moving in a preferred direction, the preferred side, or ON-SACs positioned close to the edge of the dendritic arbour first activated by null direction light stimuli, the null side (Fig. 2a,b inset and Fig. 2c,d, inset, Supplementary Fig. 2; light bar size = 100 by 300–400 μm, moved in one of 12 directions at 0.24 mm s$^{-1}$; direction selectivity index (DSI) = 0.93 ± 0.03; SAC-DSGC soma separation = 270 ± 14 μm; $n = 17$). When the somata of ON-SACs were positioned close to the edge of the preferred side of ON-DSGCs, and the sites of close dendro-dendritic SAC-DSGC apposition restricted to the preferred side, net excitatory responses were generated (Fig. 2a,b,e; PSP amplitude = 2.02 ± 0.37 mV, integral = 31.4 ± 4.3 μV.s; $n = 6$ pairs). In contrast, when the somata and sites of dendro-dendritic apposition were focussed on the null side, responses were on average inhibitory, but demonstrated a wide range (Fig. 2c-e; PSP amplitude = − 0.59 ± 0.8 mV; integral = − 33.7 ± 21.0 μV.s; $n = 11$ pairs; significantly different from preferred side: $P = 0.035$, T = 2.31; integral: $P = 0.041$, T = 2.24). Under current-clamp recording conditions, therefore,

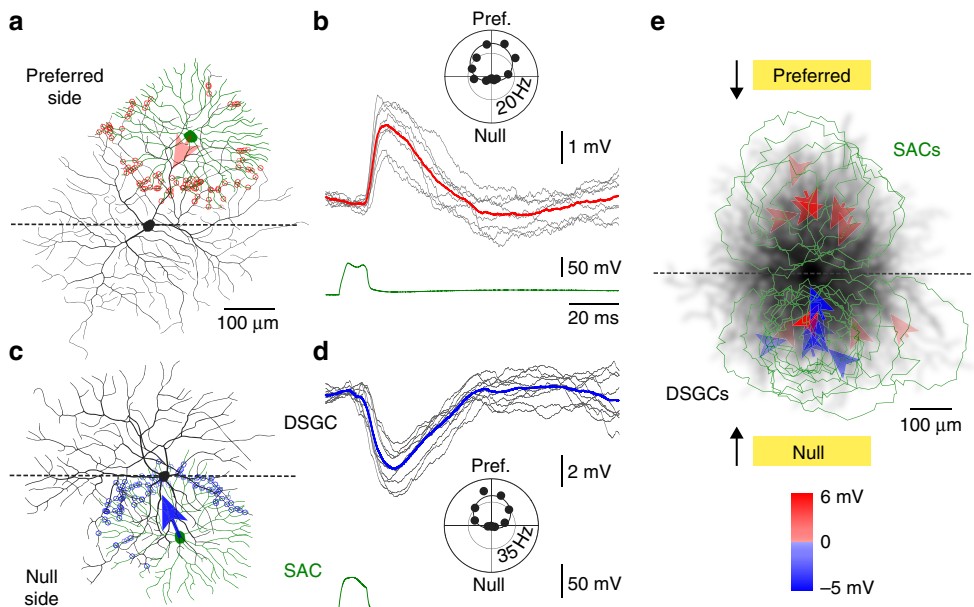

**Figure 2 | The impact of ON-SACs aligns with the directional tuning of postsynaptic ON-DSGCs. (a)** Reconstruction of a connected pair of ON-SAC (green) and ON-DSGCs (black), showing the sites (red symbols), and vectorial angle (red arrow, relative to SAC somata) of close dendro-dendritic appositions. Morphologies have been aligned to the preferred direction vector of ON-DSGC light responses (**b** inset). In all panels the preferred side of the ON-DSGC, the dendritic field first activated by light stimuli moving in the preferred direction, lies above the horizontal dashed line. Note that the SAC somata and all dendro-dendritic close appositions are positioned on the preferred side of the postsynaptic ON-DSGC. (**b**) Paired recordings from the illustrated cells (**a**) revealed that activation of the SAC-evoked excitatory PSPs in the ON-DSGC (grey traces, red trace is a digital average). The somatic current-evoked presynaptic SAC voltage responses are shown below (green trace). (**c**) Reconstruction of an ON-SAC located in the null side of an ON-DSGC. The morphologies have been aligned to the ON-DSGC light response vector (**d** inset). (**d**) Paired recordings from the illustrated cells (**c**) revealed SAC-mediated inhibitory PSPs (grey traces, blue trace is a digital average). (**e**) Summary of the impact of SACs located on the preferred and null sides of postsynaptic ON-DSGCs. The morphologies of postsynaptic ON-DSGCs are shown as overlain and spatially filtered (50 μm, black) reconstructions registered to the preferred direction of light responses, shown schematically by the yellow light bars. Nested within this field are perimeter representations of SACs (green). Arrows indicate the vector angle of SAC-DSGC close dendro-dendritic appositions coloured according to their postsynaptic impact (red excitatory, blue inhibitory PSPs). Vector length represents 1 contact per μm. All light stimuli, the direction of which are schematically illustrated, were applied under photopic conditions. Stimulus intensity was twice that of background illumination.

the balance of SAC-mediated cholinergic excitation outweighs that of inhibition on the preferred side of ON-DSGCs, while inhibition outweighs excitation on the null side (Fig. 2e). This finding parallels the targeted dendritic impact of SAC-mediated excitation and inhibition reported for ON–OFF DSGCs[16,26,27,36], and so reveals stereo-typed dendro-dendritic circuitry in the ON- and OFF-sublamina of the inner plexiform layer.

**Cholinergic excitation controls light-evoked neuronal output.** To explore the contribution of SAC-mediated cholinergic excitation to the light-evoked action potential output of DSGCs we presented moving light bar stimuli that were swept across the receptive fields of ON-DSGCs (Fig. 3; Supplementary Fig. 3). Antagonism of nAChRs dramatically reduced the robust action potential firing of ON-DSGCs evoked by light bars moved in the preferred direction and virtually eliminated the sparse action potential firing evoked by null direction stimuli, an effect evident in somatic whole-cell and cell-attached recording modes (Fig. 3a–c; Supplementary Fig. 3; preferred direction: control $= 27.4 \pm 2.5$ Hz; nAChR antagonist $= 3.1 \pm 0.8$ Hz; $P < 0.0001$, T $= 11.62$; null direction: control $= 1.1 \pm 0.3$ Hz; nAChR antagonist $= 0.03 \pm 0.02$ Hz; $P = 0.0008$, T $= 3.77$; $n = 28$). Analysis of whole-cell recorded light responses revealed that nAChR antagonists significantly reduced the membrane depolarization evoked by preferred direction light stimuli, and transformed null direction responses from weakly excitatory to inhibitory (Fig. 3a). This effect could not be accounted for by the reduction of action potential firing, as this transformation was

apparent when action potentials were digitally removed using a median filter (10 ms)[46] (Fig. 3d; Supplementary Fig. 4; voltage integral of median filtered light responses: preferred direction: control $= 12.0 \pm 1.0$ mV.s; nAChR antagonist $= 0.6 \pm 0.7$ mV.s; $P < 0.0001$, T $= 11.2$; null direction: control $= 4.0 \pm 1.1$ mV.s; nAChR antagonist $= -8.7 \pm 0.7$ mV.s; $P < 0.0001$, T $= 10.27$; $n = 28$). Furthermore, when directional tuning was disrupted by antagonism of $GABA_A$ receptors, the blockade of nAChRs powerfully decreased moving light bar-evoked action potential output in all tested directions (Supplementary Fig. 5; action potential firing rate: GABAzine $= 35.8 \pm 3.6$ Hz, GABAzine + Hex $= 7.8 \pm 3.8$ Hz; $P < 0.0001$, T $= 10.96$; $n = 6$; DSI: GABAzine $= -0.03 \pm 0.03$, GABAzine + Hex $= -0.14 \pm 0.05$).

To further explore the role of cholinergic signalling in the control of the light-evoked action potential output of DSGCs we disrupted the hydrolysis of ACh by the application of the acetylcholinesterase (AChE) inhibitor ambenonium. In the presence of ambenonium the action potential firing of ON-DSGCs evoked by preferred and null direction light stimuli were powerfully facilitated and directional-selectivity disrupted (Fig. 4a,b; DSI: control $= 0.93 \pm 0.01$; ambenonium $= 0.17 \pm 0.03$; $P = 0.002$, Wilcoxon signed rank test; voltage integral of median filtered light responses: preferred direction: control $= 8.8 \pm 0.8$ mV.s; ambenonium $= 27.6 \pm 2.1$ mV.s; $P = 0.0002$, T $= 9.85$; null direction: control $= 2.2 \pm 0.6$ mV.s; ambenonium $= 28.8 \pm 3.1$ mV.s; $P = 0.0004$, T $= 8.17$; $n = 6$). Previous work has demonstrated that the pharmacological manipulation of AChE allows investigation of the spatial relationship between

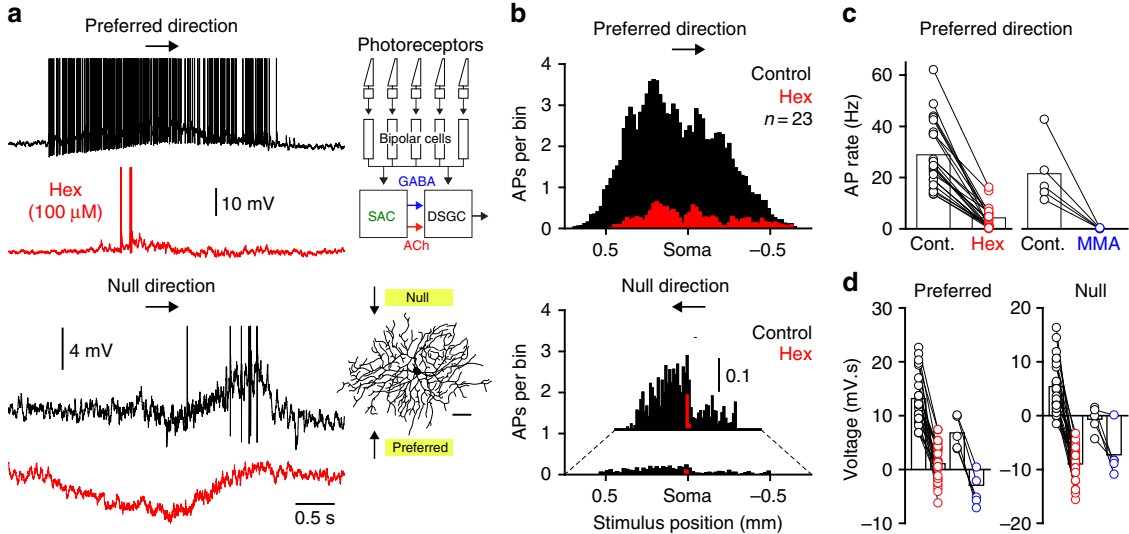

**Figure 3 | Cholinergic signalling powerfully controls the light-evoked action potential output of ON-DSGCs.** (**a**) Physiological activation of the retinal microcircuit (summarized in the inset schematic) by light bars moved across the receptive field of an ON-DSGCs (inset morphological reconstruction) leads to the generation of powerful action potential (AP) firing when moved in a preferred direction, but sparse AP output when moved in a null direction (control, black traces, APs have been truncated for clarity). The antagonism of nAChRs leads to a reduction of preferred direction light-evoked AP firing and the generation of pure inhibitory responses when light stimuli are moved in the null direction (hexamethonium (Hex); red traces). (**b**) Peri-stimulus histogram of preferred and null direction light-evoked AP firing under the indicated conditions (bin size 20 μm). (**c**) Quantification of the reduction of preferred direction light-evoked AP firing by nAChR antagonists (mecamylamine (MMA); blue symbols). (**d**) Quantification of the voltage integral of median filtered (10 ms) preferred and null direction light responses under the indicated conditions (control versus Hex: preferred: $P < 0.0001$, $T = 10.19$; null: $P < 0.0001$, $T = 11.47$; control versus MMA: preferred: $P < 0.0001$, $T = 15.86$; null: $P = 0.0038$, $T = 6.02$). All light stimuli were applied under photopic conditions. Stimulus intensity was twice that of background illumination.

cholinergic release sites and postsynaptic nAChRs, finding that manipulation of ACh hydrolysis alters cholinergic signalling only when release sites are spatially distant to activated postsynaptic nAChRs[47,48]. We therefore examined if blockade of AChE controlled pharmacologically isolated unitary nAChR-mediated PSPs evoked in paired SAC-DSGC recordings. Under these conditions the application of the AChE inhibitor ambenonium significantly enhanced unitary cholinergic excitatory transmission, but did not affect pharmacologically isolated unitary GABAergic inhibition (Fig. 4c,d; excitatory PSP integral: control = 20.8 ± 4.2 μV.s; ambenonium = 54.3 ± 11.5 μV.s; $P = 0.011$, $T = 4.49$; $n = 5$). Furthermore paired SAC-DSGC recording revealed that the augmentation of ACh hydrolysis by the local application of exogenous AChE reversibly attenuated the amplitude of pharmacologically isolated nAChR-mediated PSPs (Supplementary Fig. 6; AChE (0.4 U per μl) dissolved in Ames solution: control = 1.87 ± 0.25 mV; AChE puff = 1.10 ± 0.17 mV; $P = 0.0023$, $T = 5.72$; $n = 6$). In contrast, the control local application of Ames solution did not alter SAC-DSGC excitatory transmission (Supplementary Fig. 6; control = 2.15 ± 0.64 mV; Ames puff = 1.99 ± 0.67 mV; $P = 0.105$, $T = 2.09$; $n = 5$). The bi-directional control of unitary SAC-evoked cholinergic transmission by the augmentation and reduction of AChE activity is therefore consistent with a spatial separation between SAC release sites and activated postsynaptic AChRs[47,48]. To verify that ACh release controls preferred and null direction light responses we depleted presynaptic ACh by blocking the vesicular ACh transporter with vesamicol[49]. When presynaptic ACh release was depleted, preferred and null direction light responses were severely attenuated, and SAC-DSGC excitatory synaptic transmission selectively depressed (Fig. 4e–h; voltage integral of median filtered light responses: preferred direction: control = 14.5 ± 3.4 mV.s; vesamicol = -3.3 ± 1.5 mV.s; $P = 0.0008$, $T = 9.05$; null direction: control = 4.9 ± 3.2 mV.s; vesamicol = -11.1 ± 1.9 mV.s; $P = 0.0004$, $T = 11.11$; $n = 5$; excitatory PSP amplitude: control = 3.0 ± 0.6 mV;

vesamicol = 0.9 ± 0.2 mV; $P = 0.004$, $T = 4.47$; $n = 7$). Taken together these data reveal that ACh release from ON-SACs powerfully controls the physiological responsiveness of ON-DSGCs through the activation of postsynaptic nAChRs, in a manner consistent with a local paracrine form of neurotransmission.

To examine if cholinergic excitation influenced the responsiveness of ON-DSGCs to light stimuli across a wide stimulus range, we systematically varied the intensity of light stimuli and their speed of motion to generate stimulus-response relationship under control conditions and in the presence of a nAChR antagonist (Fig. 5a,b; intensity: 10 to 100% above background; speed: 0.04–0.9 mm s$^{-1}$). Under control conditions the directional selective output responses of ON-DSGCs emerged at the light stimulus threshold for action potential firing and were maintained across a 50-fold preferred direction firing range (Fig. 5c). The antagonism of nAChRs increased the threshold light intensity, constrained the speed of stimuli that generated action potential output, and attenuated action potential firing throughout the range of light stimuli (Fig. 5a,b). Notably, this compression of dynamic range was not accompanied by a degradation of the fidelity of the computation of direction selectivity (Fig. 5c; DSI: control = 0.946 ± 0.014; Hex = 0.990 ± 0.004; $n = 70$ trials, $n = 9$ cells). To further explore the light range over which SAC-mediated cholinergic signalling controlled the responsiveness of ON-DSGCs, we made recordings in preparations adapted to mesopic light conditions (background illumination and stimulus intensity reduced to 0.06 of the standard photopic levels). Under these dim light conditions antagonism of nAChRs dramatically reduced the action potential output of ON-DSGCs evoked by preferred and null direction light bars (Supplementary Fig. 7; preferred direction: control = 15.1 ± 2.3 Hz; nAChR antagonist = 0.4 ± 0.1 Hz; $P < 0.0001$, $T = 6.314$; $n = 11$; null direction: control = 1.0 ± 0.3 Hz; nAChR antagonist = 0.04 ± 0.02 Hz; $P = 0.0027$, $T = 3.961$). Altogether

**Figure 4 | Selective pre or postsynaptic manipulation of cholinergic signalling controls light responses.** (**a**) Blockade of acetylcholinesterase activity powerfully augments action potential (AP) firing evoked by preferred and null direction light bars (APs have been truncated for clarity). (**b**) Peri-stimulus histograms of light-evoked AP firing under the indicated conditions (ambenonium (ABN); bin size 20 μm). (**c**) Ambenonium augments pharmacologically isolated SAC-mediated excitatory PSPs (recorded in tetrodotoxin (TTX); 1 μM and GABAzine (10 μM)), but not inhibitory PSPs (recorded in TTX and mecamylamine (MMA); 10 μM). (**d**) Pooled data showing the selective augmentation of the area of excitatory PSPs by the blockade of acetylcholinesterase activity (area measured over a 100 ms time-window). (**e**) Blockade of the vesicular ACh transporter attenuates preferred direction light-evoked AP output, and unmasks null direction membrane hyperpolarization. (**f**) Peri-stimulus histograms of light-evoked AP firing under the indicated conditions (vesamicol (VES); bin size 20 μm). (**g**) Vesamicol attenuates pharmacologically isolated SAC-mediated excitatory PSPs, but not inhibitory PSPs. (**h**) Pooled data showing the selective reduction of excitatory PSPs by the pharmacological blockade of the vesicular ACh transporter (PSP amplitude has been normalized). All light stimuli were applied under photopic conditions. Stimulus intensity was twice that of background illumination. Data in **d,h** represent mean ± s.e.m.

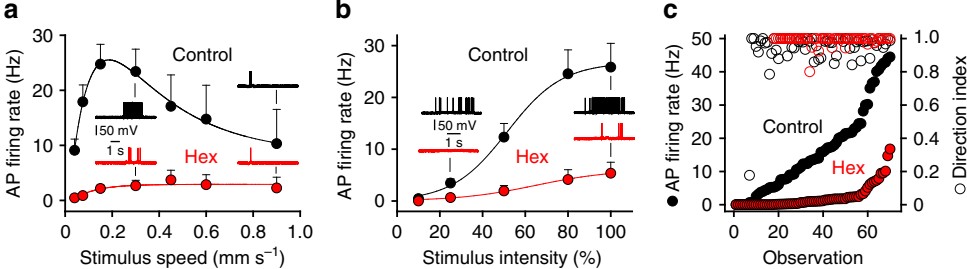

**Figure 5 | Cholinergic signalling controls the stimulus sensitivity and dynamic range of ON-DSGCs.** (**a,b**) Antagonism of nAChRs (hexamethonium (Hex); red symbols and traces) attenuates action potential (AP) firing evoked by preferred direction light bars moved across the receptive field over a wide-range of speeds (**a**), and at the optimal speed across a range of light intensities (**b**; speed = 0.24 mm/s). (**c**) Impact of cholinergic signalling on the computation of direction selectivity, and preferred direction light-evoked AP firing over the stimulus range shown in **a,b**. Note the decreased stimulus sensitivity, attenuation of AP firing rate, but preservation of directional tuning in hexamethonium. All light stimuli were applied under photopic conditions. Stimulus intensity is relative to background illumination.

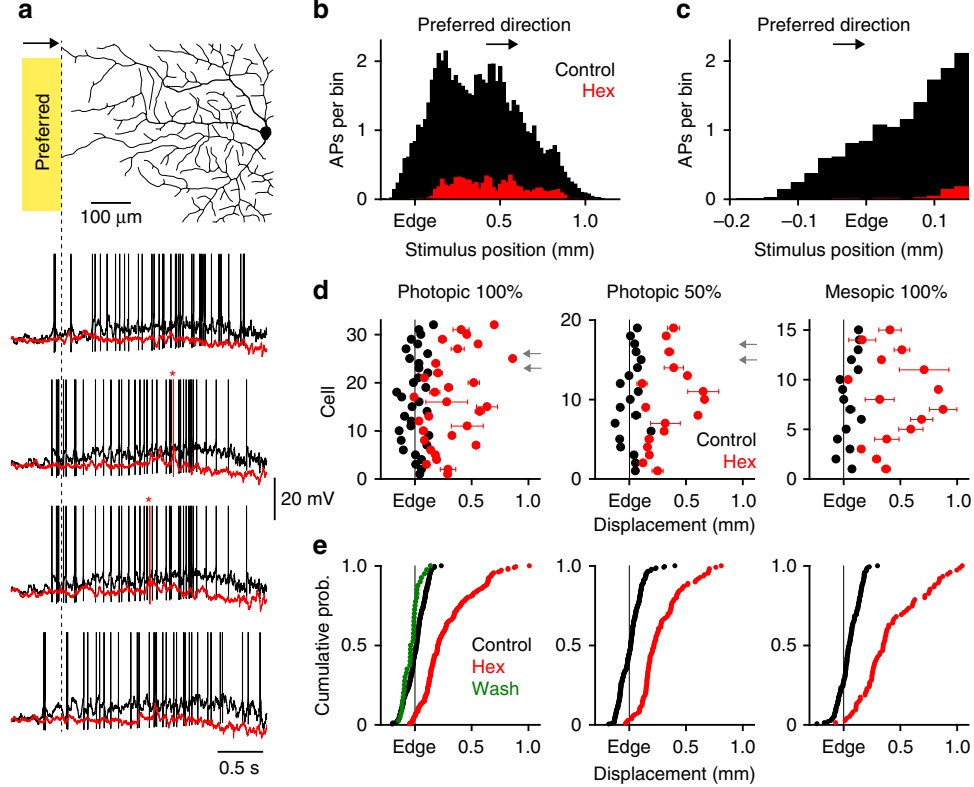

**Figure 6 | Cholinergic signalling controls receptive field structure.** (**a**) The tight spatial relationship between the onset of action potential (AP) firing and the peripheral edge of the dendritic arbour of ON-DSGCs is disrupted by nAChR antagonism (control, black traces; hexamethonium (Hex); 100 μM, red traces, * indicates first AP). The dashed reference line shows the edge of the partially reconstructed dendritic tree. Light stimuli were applied under photopic conditions (stimulus intensity 50% greater than background illumination). (**b,c**) Dendritic edge aligned peri-stimulus histogram of preferred direction light-evoked AP firing under control (black) and in the presence of nAChR antagonists (red; Hex; 100 μM, bin size 20 μm, photopic 50% stimuli). (**d**) Cholinergic excitation controls receptive field size across a wide range of light intensities. Graphs summarize the relationship between the mean ( ± s.e.m.) spatial position of the first AP evoked by preferred direction light stimuli relative to the edge of the preferred dendritic subtree under control (black) and in nAChR antagonists (red). The percentage stimulus intensity, relative to background, is shown above each graph. Each row represents the results from a single cell ( > = 5 trials under each condition), the grey arrows indicate ON-DSGCs that did not generate AP output in the presence of the nAChR antagonist. (**e**) Cumulative probability plots of the spatial position of the first AP evoked by preferred direction light stimuli relative to the edge of the preferred dendritic subtree in single trials, data are illustrated over the stimulus intensity range indicated in **d**.

these data show that cholinergic excitation acts to control the light stimulus-sensitivity and the magnitude of ON-DSGC output responses.

**Cholinergic signalling drives active dendritic integration.** To gain insight into how SAC-mediated cholinergic excitation is integrated in postsynaptic ganglion cells we first determined if cholinergic signalling controlled the receptive field structure of ON-DSGCs. Under control conditions we found a tight spatial relationship between the size of the supra-threshold receptive field and the dendritic field size of ON-DSGCs across a wide-range of light conditions, consistent with previous reports[50] (Fig. 6; receptive field size = 923 ± 24 μm; dendritic field size = 851 ± 23 μm, $r = 0.574$; $P < 0.0001$, $n = 66$ cells). Notably, when preferred direction light bars were swept across the receptive field a close relationship was found between the time at which light stimuli activated the retinal circuitry which innervated the edge of the preferred dendritic subfield of ON-DSGCs and the time of occurrence of the first light-evoked action potential, which could be converted into a spatial relationship (Fig. 6a–e; control: displacement = 19.8 ± 3.6 μm; $n = 642$ trials, $n = 66$ cells). The pharmacological blockade of

nAChRs reduced action potential firing and delayed its onset as preferred direction light bars were swept across the receptive field, resulting in a disruption of the relationship between dendritic field edge and action potential generation, and a dramatic constriction of supra-threshold receptive field size (Fig. 6a–e; control = 923 ± 24 μm; nAChR antagonist = 471 ± 40 μm; $P < 0.0001$; T = 11.27; $n = 66$). The blockade of SAC-mediated cholinergic excitation disrupted the wide field receptive field properties of ON-DSGCs to a similar degree under photopic and mesopic light conditions (compare graphs in Fig. 6d,e), suggesting that SAC-mediated cholinergic signalling functions to powerful drive dendritic excitation across a broad physiological range of stimuli.

Previous observations from the rabbit retina have indicated that active dendritic integration is essential for the emergence of the wide receptive field properties of ON-DSGCs[43,45]. We therefore used simultaneous somato-dendritic recording techniques to directly examine the dendritic mechanisms underlying the integration of SAC-mediated cholinergic excitation. Under control conditions simultaneous somato-dendritic recordings demonstrated that preferred direction light stimuli drove the robust generation of dendritic spikes, which forward propagated to the soma and axon to initiate

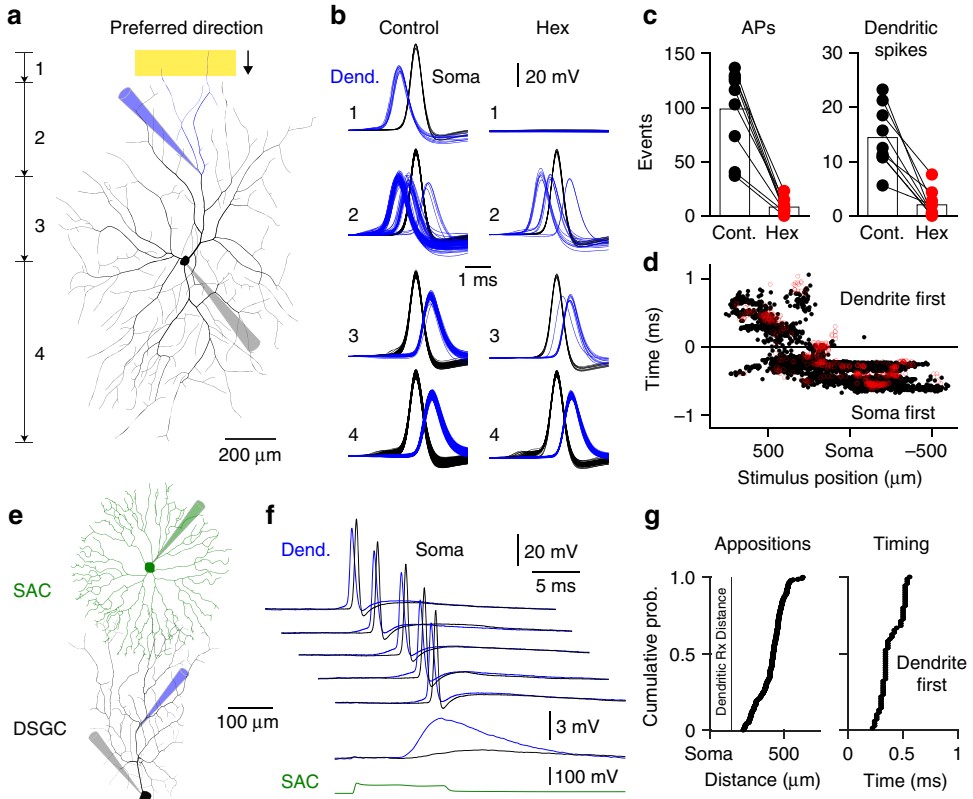

**Figure 7 | Cholinergic signalling controls light-evoked dendritic spike initiation.** (**a**) Reconstruction of an ON-DSGC showing the placement of recording electrodes and the preferred direction movement of a light bar; the blue coloured section of the dendritic tree feeds to the dendritic recording site. (**b**) The generation of dendritic spikes (dendritic recording (blue traces), positions 1 and 2 in **a**), and back-propagating action potentials (positions 3 and 4 in **a**) were attenuated by antagonism of nAChRs (hexamethonium (Hex); 100 μM) when a preferred direction light bar was swept across the receptive field. Traces are aligned to the peak of somatically recorded action potentials (APs). (**c**) Summary of the reduction of AP output and dendritic spike generation by hexamethonium. (**d**) Spatial pattern of somato-dendritic spike delay under control (black symbols) and following the blockade of nAChRs (Hex; red symbols). (**e**) Reconstruction of simultaneously recorded ON-SAC and ON-DSGCs, showing the placement of recording electrodes. (**f**) ON-DSGC somatic (black traces) and dendritic (blue traces) recordings show that the presynaptic SAC drives dendritic spike generation (reconstruction in **a**; recordings were made in GABAzine (10 μM); five consecutive SAC-evoked supra-threshold excitatory responses are shown). The lower overlain traces show the dendro-somatic attenuation of a sub-threshold voltage response evoked by SAC activation (green trace). (**g**) Quantification of the sites of SAC-DSGC close apposition in recorded DSGC dendritic subtrees (left graph, vertical line represents average dendritic recording site; 145 ± 5 μm from the soma; n = 5). Note that all detected appositions were distal to dendritic recording sites. The right graph shows that dendritic spikes preceded action potentials in all supra-threshold trials.

action potential firing (Fig. 7a,b; Supplementary Fig. 8). The pharmacological blockade of SAC-mediated cholinergic excitation dramatically attenuated light-evoked dendritic spike generation and consequential action potential output (Fig. 7a–d). Analysis revealed that cholinergic excitation controlled the number of dendritic spikes evoked by light bars as they excited terminal dendritic sites which fed to the site of dendritic recording, illustrating that SAC-mediated cholinergic excitation is integrated locally in the dendritic arbour of ON-DSGCs (Fig. 7a–d, terminal dendritic tree highlighted in blue in the reconstruction; dendritic spikes per episode: control = 14.4 ± 1.9; Hex = 2.0 ± 0.8; P = 0.0002, T = 6.4; n = 9; dendritic recording preferred side = 275 ± 22 μm from soma; n = 5; null subfield = 265 ± 41 μm; n = 4, Supplementary Fig. 8). To explore the determinants of this form of sub-cellular integration, we tested if cholinergic excitation provided by a single SAC was capable of driving the initiation of dendritic spikes in ON-DSGCs. To do this we made simultaneous somato-dendritic recordings from ON-DSGCs and a third somatic recording from a presynaptic ON-SAC, in the presence of the GABA$_A$ receptor antagonist GABAzine

(Fig. 7e–g; n = 7). Under these conditions the activation of a single SAC led to the generation of nAChR-mediated dendritic excitatory PSPs, which were crowned by the firing of dendritic spikes, in recordings where dendro-dendritic SAC-DSGC appositions were focussed in the recorded dendritic arbour of the ON-DSGCs at loci distal to the site of dendritic recording (Fig. 7e–g; dendritic recordings 145 ± 5 μm from soma; average site of SAC dendro-dendritic appositions from DSGC soma = 437 ± 26 μm; n = 5). Simultaneous somatic recording revealed that each dendritic spike preceded and drove action potential firing (dendritic spike to action potential delay = 0.40 ± 0.05 ms). In contrast, when dendritic recordings were made from the dendritic subfield contralateral to the site of predominate SAC innervation, SAC-evoked cholinergic excitation drove action potential firing, which was first recorded somatically and subsequently back-propagated to the dendritic recording site, consistent with cholinergic excitation of the contralateral dendritic tree (Supplementary Fig. 9). Taken together these data directly demonstrate that dendro-dendritic cholinergic excitation is capable of driving the generation of dendritic spikes in ON-DSGCs.

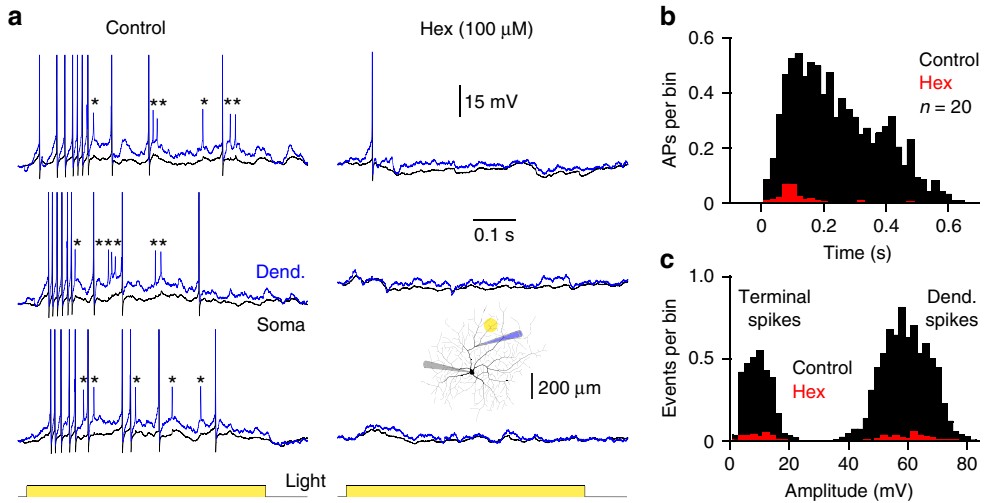

**Figure 8 | Cholinergic signalling drives a cascade of dendritic spike generators. (a)** Dramatic attenuation of light spot-evoked terminal dendritic (marked by *) and large-amplitude dendritic spike generation by the blockade of nAChRs (Hex; 100 μM). The inset shows ON-DSGC morphology, and the placement of recording electrodes and light spot stimuli. **(b)** Peri-stimulus time histogram of AP firing evoked by dendritic light spots (ON time = 0–0.5 s) under the indicated conditions. **(c)** Amplitude distributions of light spot-evoked dendritic regenerative activity under control (black) and in hexamethonium show the nAChR-mediated control of terminal and large-amplitude dendritic spike generation.

**Cholinergic signalling gates dendritic spike initiation.** To quantitatively examine how light-evoked dendro-dendritic cholinergic excitation controls the generation of dendritic spikes, we flashed light spots at terminal dendritic sites under control and in the presence of nAChR antagonists (Fig. 8; light spot = 105 ± 5.2 μm; n = 20). In confirmation of previous observations[45], direct dendritic recordings revealed that light spots evoked both small- and large-amplitude dendritic spikes under control conditions. Small-amplitude, terminal dendritic, spikes were compartmentalized in the dendritic tree and acted as variable triggers of large-amplitude dendritic spikes, which preceded and drove action potential output (Fig. 8a). The application of nAChR antagonists significantly limited the initiation of dendritic spikes, by decreasing the occurrence of light spot-evoked terminal and large-amplitude dendritic spikes, and as a consequence action potential output (Fig. 8a–c). In support of the central role of cholinergic excitation in this form of local dendritic integration, simultaneous somato-dendritic recordings revealed that the dendritic depolarization evoked by light spot stimuli was dramatically reduced in amplitude by the antagonism of nAChRs when recordings were made in the presence of tetrodotoxin (TTX) and GABAzine to block dendritic spike generation and SAC-mediated inhibition, respectively[45] (Supplementary Fig. 10; light spot = 100 μm; dendritic recordings 240 ± 11 μm from soma; n = 7, control: dendrite = 16.6 ± 2.2 mV, soma = 4.5 ± 0.7 mV; Hex: dendrite = 6.8 ± 1.2 mV, soma = 1.7 ± 0.3 mV; P = 0.0008, T = 6.24 and P = 0.0009, T = 6.04, respectively; n = 7). Taken together, these data reveal that light spot-evoked bipolar-cell-mediated glutamatergic[12,42] and SAC-mediated cholinergic excitation are integrated locally in the dendritic tree to influence the initiation of terminal dendritic spikes, and the subsequent recruitment of large-amplitude parent dendritic spikes, which in turn, drive action potential output.

To critically evaluate the contribution of cholinergic excitation in this multi-level integrative process we attempted to recreate the postsynaptic effects of light-evoked dendritic cholinergic excitation on dendritic spike initiation when nAChRs were pharmacologically blocked, or when presynaptic ACh release was depleted (Fig. 9). When nAChRs were blocked, light spot-evoked terminal and large-amplitude dendritic spike generation was significantly attenuated. The generation of dendritic spikes by light-evoked bipolar cell-mediated glutamatergic excitatory input could however be rescued by the pairing of light spot stimuli with direct dendritic depolarization generated by the injection of sub-threshold steps of positive current through the dendritic recording electrode (Fig. 9a,b; light spot = 107.1 ± 12 μm; dendritic current injection = 138.1 ± 27.5 pA; dendritic recordings 215 ± 12 μm from soma; n = 7). Notably, the occurrence of both terminal and large-amplitude dendritic spikes were significantly augmented by the pairing of light spot stimuli with direct dendritic depolarization, whereas dendritic depolarization alone failed to evoke small-amplitude, terminal dendritic spike generation, but infrequently reached the threshold for the generation of large-amplitude dendritic spikes in a fraction of trials (Fig. 9c,d; occurrence of terminal and large-amplitude dendritic spikes significantly different between paired and unpaired Hex conditions; P = 0.0071, T = 4.0 and P = 0.0274, T = 2.9, respectively; n = 7). In order to more closely replicate the actions of light-evoked ACh release from SACs we paired the local dendritic iontophoretic delivery of ACh with light spot excitation, when SAC-mediated ACh release had been depleted by blocking the ACh vesicular transporter (Fig. 9e,f; Supplementary Fig. 11). Under these conditions the pairing of ACh iontophoresis with light spot stimuli powerfully drove dendritic spike generation and action potential output, whereas light stimuli, or ACh iontophoresis alone infrequently reached the threshold for dendritic spike generation (Fig. 9e,f; occurrence of terminal and large-amplitude dendritic spikes significantly increased in the paired condition; P = 0.0047; q = 6.44 (light alone) and P = 0.0012; q = 7.99 (iontophoresis alone); ANOVA; n = 5; dendritic recordings 315 ± 15 μm from soma). Consistent with this, when SAC-mediated excitation and inhibition were blocked, the pairing of light-evoked bipolar cell-mediated excitatory input with direct dendritic depolarization dramatically facilitated dendritic spike generation and neuronal output, underscoring the role of SAC-mediated inhibition in the control of dendritic spike generation[45] (Supplementary Fig. 12; light spot = 87.5 ± 12.5 μm; dendritic recordings 132 ± 14 μm from soma; dendritic current = 225 ± 47.1 pA; dendritic spikes per episode: unpaired = 1.1 ± 0.5; paired = 12.8 ± 2.9; P = 0.003, T = 6.3; n = 4). These data therefore demonstrate

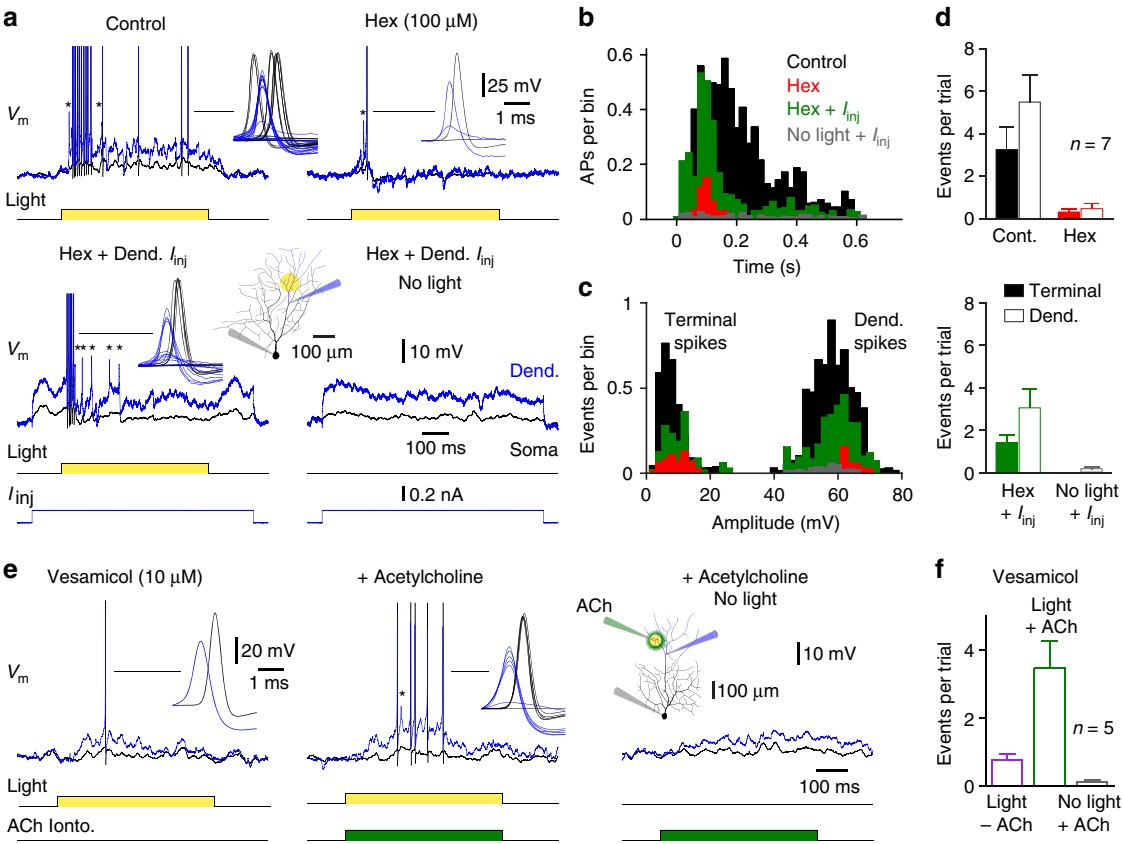

**Figure 9 | Cholinergic signalling positively gates terminal dendritic spike generation.** (**a**) Light spot-evoked terminal dendritic (marked by *) and large-amplitude dendritic spike generation were powerfully attenuated by nAChR antagonism (upper traces, hexamethonium; Hex). This reduction could be rescued by pairing light stimuli with sub-threshold dendritic depolarization (lower left traces). The insets show light spot-evoked dendritic spikes (blue traces) and somatically recorded activity (black traces). A section of the ON-DSGC morphology, the placement of electrodes, and the position of the light spot stimuli are also shown. (**b**) Peri-stimulus time histogram of light spot-evoked (ON time = 0–0.5 s) action potential (AP) output under the indicated conditions. (**c**) Amplitude distribution of dendritically recorded spikes under the indicated conditions. Note that pairing of light spot stimuli with dendritic depolarization increased the generation of both terminal and large-amplitude dendritic spikes. (**d**) Number of terminal and large-amplitude dendritic spikes evoked by light spot stimuli under the indicated conditions (data represent mean ± s.e.m). (**e**) When presynaptic ACh release was depleted (vesamicol), the pairing of light spot stimuli with the local dendritic iontophoretic delivery of ACh augmented dendritic spike generation. The insets show overlain spikes, a section of the ON-DSGC morphology, the placement of recording and iontophoresis electrodes, and the position of light spot stimuli. (**f**) Number of terminal and large-amplitude dendritic spikes evoked by light spot stimuli per trial under the indicated conditions (data represent mean ± s.e.m).

that SAC-mediated cholinergic and bipolar cell-mediated glutamatergic excitation drive the generation of dendritic spikes, revealing a mechanism by which dendro-dendritic ACh release acts in a spatially selective manner in the dendritic tree of DSGCs to control neuronal output.

## Discussion

The retinal microcircuitry that underlies the processing of image motion has been solved at ultrastructural resolution[9,17], but a full understanding of the mechanisms underlying the computation of direction selectivity requires investigation of circuit function at a similar scale. Here we used multi-site somatic and dendritic electrophysiological recording techniques to demonstrate that the direction-selective action potential output of rabbit ON-DSGCs is powerfully controlled by SAC-mediated dendro-dendritic cholinergic excitation.

In confirmation of previous findings obtained from ON-OFF-DSGCs[16,26,34–36], our paired ON-SAC to ON-DSGC recordings revealed the dendro-dendritic co-release of ACh and GABA from SACs, which drove postsynaptic excitation and inhibition, mediated by the activation of nAChR and GABA_A receptors, respectively. Notably, we observed that the

postsynaptic impact of SACs was defined by their position in the receptive field of postsynaptic DSGCs. SACs positioned on the preferred side, which made close dendro-dendritic appositions from dendrites oriented in the preferred direction of DSGCs, generated net excitatory responses in DSGCs, whereas, SACs located on the null side generated, on average, postsynaptic inhibition. These data suggest that when light stimuli enter the receptive fields of DSGCs in a preferred or null direction, the activation of SACs located on the periphery of the preferred and null side should differentially influence the action potential output of ON-DSGCs. Consistent with this, when the direction-selective microcircuit was physiologically activated, the robust preferred direction light-evoked action potential output of ON-DSGCs was greatly attenuated by the pharmacological antagonism of nAChRs, and by the selective depletion of presynaptic ACh release from SACs. Notably, we found across a broad range of light stimuli that the antagonism of nAChRs disassociated the tight relationship between the receptive and dendritic field areas of ON-DSGCs, and disrupted the characteristic generation of action potential firing evoked by preferred direction light bars as they entered the receptive field.

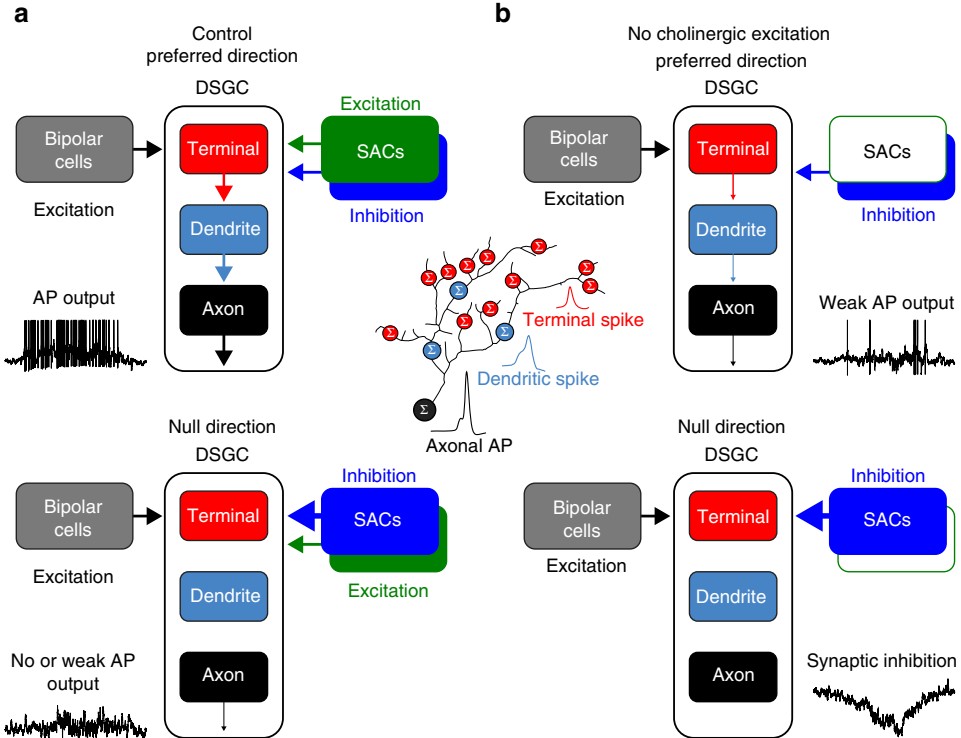

**Figure 10 | Schematic summary of the role of SAC-mediated control of active dendritic integration compartments of ON-DSGCs.** (**a**) SAC-mediated cholinergic excitation and GABAergic inhibition gate terminal dendritic integration to control action potential output under control conditions. (**b**) In the absence of cholinergic excitation terminal dendritic spike generation is weakly driven by bipolar cell-mediated excitation.

Direct dendritic electrical recording, calcium imaging and computational modelling have shown that active dendritic integration underlies the wide-field responsiveness of DSGCs[13,43–45]. Multi-site somato-dendritic recording techniques have deconstructed this process to reveal that dendritic spikes are first initiated in the small calibre terminal dendrites of DSGCs[45], which receive excitatory glutamatergic synapses from bipolar-cells[51,52]. Terminal dendritic spike generation has been found to be the first step in a multi-layered active integration cascade, as these spikes act as variable triggers for the generation of dendritic spikes in larger calibre parent dendrites that, once initiated, forward propagate to the axon to drive action potential output[45]. Our results demonstrate that SAC-mediated cholinergic signalling is a major component of this integrative process. Simultaneous somato-dendritic recordings revealed that the generation of terminal and large-amplitude parent dendritic spikes by light spots or preferred direction moving light bars were powerfully attenuated by the antagonism of nAChRs. An effect underscored by the rescue of dendritic spike generation when cholinergic excitation was mimicked by local dendritic depolarization or the terminal dendritic application of ACh. The dendro-dendritic release of ACh therefore controls the action potential output of DSGCs by driving terminal dendritic spike generation, and enhancing the coupling between active dendritic integration sites. We suggest that SAC-mediated cholinergic excitation acts as a dendritic subunit-specific gate of bipolar-cell signalling, that when activated by moving light bars powerfully controls the action potential output of DSGCs, to define the stimulus sensitivity and receptive field area over which the computation of direction selectivity is executed (Fig. 10). An idea supportive of previous observations made from ON-OFF-DSGCs, which have revealed that cholinergic excitation acts to control bipolar-cell mediated

glutamatergic excitation by supplying membrane depolarization to relieve the voltage-dependent block of NMDA receptor-mediated PSPs[41]. Our findings however extend this idea to demonstrate that these excitatory inputs act in concert to drive the initiation of terminal dendritic spikes, and thus launch a cascade of dendritic spike generation which ultimately results in the driving of action potential output, providing a substrate for the localized dendritic processing of light stimuli[3] (Fig. 10).

In contrast, SACs located on the null side of DSGCs drove net postsynaptic inhibition, a finding that is consistent with the greater $GABA_A$ receptor-mediated synaptic conductance, and distribution of dendro-dendritic synapses on the null side of DSGCs[16,17,26–28]. However, at these sites, we find that dendro-dendritic cholinergic signalling plays a functional role, as the blockade of cholinergic excitation transformed null direction light-evoked voltage responses from weakly excitatory to strongly inhibitory. Consistent with this, when GABAergic inhibition was pharmacologically blocked, the antagonism of nAChRs attenuated moving light bar-evoked action potential output in all tested directions, as has been shown for ON-OFF-DSGCs[23]. Furthermore, when the uptake of ACh was inhibited, both null and preferred direction light responses were strongly facilitated, and direction tuning disrupted. These data reveal that a balance between the postsynaptic impact of SAC neurotransmitters is essential for the emergence of directional selectivity, questioning how action potential firing is suppressed by null direction light stimuli.

Dendritic recording and neuronal modelling has demonstrated that null direction light stimuli evoke sparse action potential output in rabbit DSGCs because of the SAC-mediated inhibitory control of dendritic spike initiation[44,45]. Direct recording has demonstrated that this inhibitory control is restricted to terminal dendritic sites, as null direction synaptic inhibition does not

inhibit the initiation of large-amplitude parent dendritic spikes in DSGCs generated in response to steps of direct current[45]. When considered together with our current work, these data suggest that the dominant SAC-mediated null side inhibition acts to clamp the terminal dendritic membrane potential at a voltage sub-threshold for the generation of terminal dendritic spikes, thereby negatively gating their generation by bipolar-cell-mediated glutamatergic and SAC-mediated cholinergic excitation (Fig. 10).

A structural substrate for the dual excitatory and inhibitory roles of SACs in the retinal circuitry is well documented[16,23,26,27,29–36]. The dendrites of ON- and OFF-SACs form dense choline acetyl transferase and vesicular ACh transporter delineated fascicles in the inner plexiform layer, the passage of which are closely followed by the dendrites of DSGCs[51,53,54]. Notably this mapping is most refined for small terminal DSGC dendrites, the predominant site of SAC-DSGC synaptic connectivity[22]. Ultrastructural analysis of murine ON-OFF-DSGCs has, however, shown that the distribution of dendro-dendritic synapses is asymmetrical, with a $\sim 13$-fold greater density established by SACs positioned on the functional determined null side of ON-OFF-DSGCs[17]. Our results demonstrate in the simplified circuitry of the ON-sublamina of the inner plexiform layer that the postsynaptic impact of ON-SACs is position dependent, with cholinergic excitation dominating on the preferred, and GABAergic inhibition on the null side, consistent with previous results from ON-OFF-DSGCs[16,26–28,36]. As our paired recordings, and previous results, have revealed the obligatory co-release of ACh and GABA from SACs, these data suggest that ACh may be released from, or have postsynaptic impact at, sites other than dendro-dendritic synapses, a finding that is supported by the relatively long time to onset of cholinergic PSPs, and the effects of manipulating ACh hydrolysis[47,48]. Furthermore, direct dendritic recordings revealed that light stimuli, and direct activation of SACs drove localized nAChR-dependent dendritic excitation, which was integrated locally in the dendritic arbour to control dendritic spike initiation. Our findings are therefore consistent with a model in which ACh operates in a localized paracrine fashion in the IPL[17,41,55], at sites constrained by the co-fasciculation of SAC and DSGC dendrites. The development of new tools for the direct visualization of ACh release and diffusion are, however, required to definitively address this issue. Based on our findings we conclude that the excitatory and inhibitory neurotransmitters released from the dendrites of feed-forward interneurons in the output layer of the retina are integrated together with direct bipolar-cell-mediated glutamatergic excitation to drive and control the generation of dendritic spikes in DSGCs, the initiation of which determines the light stimulus sensitivity, receptive field size and direction selectivity of action potential output (Fig. 10).

## Methods

**Retinal preparation.** Adult pigmented or albino rabbits of either sex (Nanowie Small Animal Production Unit) were dark-adapted for 1-2 h, anesthetized with ketamine ($12\,mg\,kg^{-1}$ of body weight, intramuscular) and xylazine ($12\,mg\,kg^{-1}$, intramuscular) then overdosed with pentobarbitone sodium ($150\,mg\,kg^{-1}$, intravenous), before the eyes were quickly enucleated, hemisected, and placed in carbogenated Ames solution ($8.8\,g\,l^{-1}$ Ames, $1.9\,g\,l^{-1}$ sodium bicarbonate, $0.75\,g\,l^{-1}$ glucose) at room temperature ($22$–$24\,°C$). Large sections of the inferior peripheral retina were separated from the sclera and pigmented epithelial layer, and stored in carbogenated Ames solution at room temperature. A single section was placed in a recording chamber perfused with Ames solution at $35$–$37\,°C$. The Animal Ethics Committee of the University of Queensland approved all procedures.

**Whole-cell recording.** Dual and triple whole-cell recordings were made from visually identified DSGCs and SACs, using infrared differential interference

contrast video microscopy, with identical current-clamp amplifiers configured in 'bridge-mode' (BVC 700A, Dagan Corporation)[45]. Voltage and current signals were low-pass filtered at DC to 10–30 kHz and sampled at 50 kHz. DSGC recording pipettes were filled with (in mM): 135 K-gluconate; 7 NaCl; 10 HEPES; 10 phosphocreatine; 2 Na$_2$-ATP; 0.3 Na-GTP; 2 MgCl$_2$ and 0.35 Alexa Fluor 594 (Molecular Probes) (pH 7.3–7.4; KOH), and had an open tip resistance of 4–5 MΩ for somatic and 15–20 MΩ for dendritic pipettes. All DSGC recordings were made from the resting membrane potential (soma $= -52.7 \pm 0.7$ mV; $n = 156$; dendrite $= -54.9 \pm 0.4$ mV; $n = 47$; $136.9 \pm 8.8\,\mu m$ from soma). SAC recording pipettes were filled with (in mM): 110 Cs-methysulphate; 5 NaCl; 10 HEPES; 10 phosphocreatine; 2 Na$_2$-ATP; 0.3 Na-GTP; 2 MgCl$_2$ and 0.35 Alexa Fluor 488 (Molecular Probes) (pH 7.3–7.4; CsOH), and had an open tip resistance of 5–7 MΩ. For SAC recordings, a period of 7–25 min was left to allow for the intracellular blockade of potassium conductances[56], before testing connectivity with DSGCs. SAC-DSGC connectivity was tested for by the generation of threshold regenerative activity evoked by the injection of short somatic positive current steps (10 ms; $0.87 \pm 0.11$ nA; $n = 42$; usually terminated by a negative current step: 0.5–0.9 s; $-0.15 \pm 0.01$ nA; inter-trial interval 20–30 s; connectivity $= 84\%$) in the presence of the glutamate receptor antagonists (6-cyano-7-nitroquinoxaline-2,3-dione; $10\,\mu M$ and DL-2-amino-5-phosphonopentanoic acid; $100\,\mu M$). No correction was made for liquid junction potentials. Drugs were typically applied by addition to the perfusion medium at determined concentrations. In some experiments ACh was delivered by iontophoresis (100 mM in iontophoresis pipette) to a site co-aligned with light spot stimuli distal ($93 \pm 16\,\mu m$) to the dendritic recording electrode (ejection current $= 50$–$200$ nA; retention current $= 5$–$20$ nA)[45,57]. Exogenous AChE (0.4 U per µl)[47] dissolved in Ames solution, or Ames solution, were applied by pressure application ($0.58 \pm 0.03$ PSI for 10 min; $n = 11$) under visual guidance from a pipette with similar characteristics as those used for somatic recording. At the termination of each whole-cell recording, the location of the recording pipettes and neuronal morphology were examined by fluorescence microscopy, and recorded using a digital camera (QImaging) or confocal microscopy (Zeiss), and reconstructed using Neurolucida software (MBF Bioscience).

**Visual stimulation and analysis.** Visual stimuli were generated using custom software (developed by W.R. Taylor (Oregon Health and Science University) and R.G. Smith (University of Pennsylvania)), and presented on a $800 \times 600$ pixel OLED screen. Images were projected and focussed on the plane of photoreceptor outer segments to physiologically activate the retinal network using a $\times 10$ water objective lens with a numerical aperture of 0.3 (Olympus)[45]. Standard light stimuli were applied under photopic conditions (background illumination was maintained above the level of rod saturation at $\approx 3.5 \times 10^{11}$ quanta cm$^{-2}$ s$^{-1}$) and the visual stimuli were standardly set at 100% of the background, in some preparations mesopic conditions were established (background and visual stimuli reduced by 0.06 with a neutral density filter). Moving light bars ($100 \times 300$–$400\,\mu m$) were standardly moved across receptive fields in 12 directions at 30° intervals, at a speed of 0.24 mm s$^{-1}$ (inter-trial interval $= 12$–$17$ s). A range of light spot stimuli (50–$200\,\mu m$) flashed (500 ms) at determined receptive field loci were also employed. Directional selectivity was quantified by calculating a directional selectivity index of the number of action potentials, or action potential firing rate, recorded in multiple trials ($n = 5 \pm 0.5$), generated by preferred and null direction stimuli, according to the equation: preferred - null/preferred + null[35]. Directionally selective output responses were represented as polar plots, and a directional vector calculated. Peri-stimulus time histograms of preferred or null direction moving light bar-evoked action potential firing were binned for each segment when the moving bar ($240\,\mu m\,s^{-1}$) traversed $20\,\mu m$ of the receptive field. The time delay between simultaneously recorded regenerative events was measured at peak amplitude. For analysis of the time delay between simultaneously recorded spikes as a function of visual stimulus position, regenerative events from each cell were binned according to their spike times for each segment where the moving bar traversed $20\,\mu m$ of the receptive field.

**Statistical analysis.** Data sets were compared using a Student's $t$-test (two-sided), analysis of variance (Tukey's multiple comparison), the Wilcoxon signed rank test or the Mann–Whitney test, with the test selected according to data structure. The statistical difference between amplitude distributions was determined with the Kolmogorov–Smirnov test. Statistical significance was accepted at $P < 0.05$, and normality of distribution established with the Kolmogorov–Smirnov test with a Dallal–Wilkson–Lillie post test. No statistical methods were used to predetermine sample sizes. Data collection and analysis were not randomized or performed blind to the conditions of the experiments. Numerical values are presented as mean ± s.e.m.

**Data availability.** The data sets generated during and/or analysed during the current study are available from the corresponding author on reasonable request.

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

## Acknowledgements

This work was supported by grants from the Australian Research Council (FT100100502), National Health and Medical Research Council (APP1082257) and the Hand Heart Pocket Foundation to S.R.W. We are grateful to Rowan Tweedale and Ben Sivyer for their constructive comments.

## Author contributions

S.R.W. conceived the project, S.R.W. and A.B. designed the experiments, A.B. and S.K.-d.C. conducted experiments, and A.B., S.K.-d.C., E.J.C.-W. and S.R.W. analysed the data. S.R.W. wrote the paper with input from all authors.

## Additional information

**Competing interests:** The authors declare no competing financial interests.

