## [Peer Review File · Nature Communications]

Reviewers' Comments:

Reviewer #1 (Remarks to the Author):

The direction selective circuit is a classic model for studying neural computations. In this circuit, starburst amacrine cells (SBACs) are critical for generating direction selectivity of On and On-Off types of direction selective ganglion cells (DSGCs). This manuscript by Brombas et al. examines the role of cholinergic excitation from SBACs in the light response of On DSGCs in the rabbit retina. They found that cholinergic inputs from SBACs significantly enhance the firing rate of On DSGCs by generating dendritic spikes in On DSGCs.

Most studies on the neural mechanisms underlying direction selectivity have focused on On-Off DSGCs, which are more numerous than On DSGCs in the retina. In contrast, few studies focused on the synaptic and cellular mechanisms of direction selectivity of On DSGCs. Therefore, the authors fill the gap by targeting On DSGCs for somatic and dendritic recordings, and directly measuring synaptic connections between SBACs and On DSGCs. They found that cholinergic excitation from SBACs powerfully contributes to action potential firing of On DSGCs, a result that echoes previous studies on On-Off DSGCs. They then went further to show that this cholinergic facilitation is due to enhanced dendritic spike generation in On DSGCs using simultaneous dendritic and somatic recordings.

Specific comments

- My major concern is focused on the results shown in Figures 1d-1h. The On DSGCs in these figures are aligned with their preferred directions (upward). Activating SBACs located in the upper side (labeled "Preferred subfield" in Figure 1d) produced a net depolarization in On DSGCs, while activation of most SBACs on the lower side (labeled "Preferred subfield" in Figure 1d) produced a hyperpolarization (Figure 1h). This is directly contradictory to all previous studies on On-Off DSGCs and On DSGCs. It is well-accepted that the strongest inhibitory inputs onto DSGCs come from SBAC dendrites that extend to the null direction of DSGCs. The authors need to address this surprising discrepancy in their manuscript.

- The Abstract and Introduction needs to be re-worked to include an updated description of the current literature on this topic.

For example, the authors claim that "it remains unknown, however, how SBAC-mediated cholinergic excitation is integrated together with bipolar-cell-mediated glutamatergic excitation and SBAC-mediated inhibition to control the action potential output of DSGCs". In fact, two

studies published earlier this year (Sethuramanujam et al, 2016; Poleg-Polsky and Diamond, 2016) addressed exactly this question, but are omitted in this manuscript. Kittila and Massey, 1995 also needs to be included in the references.

- In Discussion, page 12, the third paragraph, the "preferred-side" and "null-side" defined in this paper is opposite to those defined in previous studies. Thus, this paragraph needs to be re-written to address the discrepancy mentioned above.

Page 15, last paragraph:

"Support for these electrophysiological data is provided by ultrastructural analysis, which shows a clear distinction amongst the distribution of different classes of dendro-dendritic contacts in the dendritic arbor of ON-OFF DSGCs 17. Morphologically identified classical dendro-dendritic synapses have been shown to be focussed in the null-dendritic subtree, whereas other classes of close SBAC dendro-dendritic appositions are uniformly distributed throughout the dendritic tree of DSGCs 17, suggesting a structural correlate for the dendritic-field dependent balance of GABAergic and cholinergic signalling."

These sentences sound vague to me: what did the authors mean by "the different classes of dendro-dendritic contacts"? Which cell type does "the null-dendritic subtree" refer to? Reference #17 by Briggman and colleagues demonstrates an antiparallel relationship between the orientation of presynaptic SBAC dendrites and the preferred direction of On-Off DSGCs, which is not consistent with the wiring pattern shown in Figure 1 of this paper.

Reviewer #2 (Remarks to the Author):

In their study, Brombas et al. show that cholinergic input from starburst amacrine cells (SACs) gates the initiation of dendritic spikes in ON direction-selective ganglion cells (oDSGCs) in the rabbit retina. Direction-selective circuits in the retina have been extensively studied. Most attention has focused on the asymmetric inhibitory input from SACs to DSGCs, which is required for direction selectivity. By comparison, less is known about the excitatory cholinergic input from SACs and the dendritic integration of both inputs by DSGCs. The present work explores these topics and reveals that cholinergic input from single SACs in the preferred subfield is sufficient to elicit spikes in oDSGCs. These spikes are blocked by cholinergic antagonists. The authors also show that pre- and postsynaptic interference with cholinergic transmission reduces oDSGC responses to visual motion stimuli and shrinks their suprathreshold receptive field. They then use direct dendritic recordings from oDSGCs to demonstrate that the effects of cholinergic transmission on somatic spiking are secondary to effects on dendritic spiking. In particular, electrical stimulation of single SACs in the preferred subfield or stimulation with small spots of light elicit dendritic spikes, which propagate to the soma of

oDSGCs. Blockers of cholinergic transmission block these dendritic spikes as well as their somatic counterparts. Finally, the authors use local manipulations of cholinergic transmission, light stimuli, and dendritic current injections to show that acetylcholine gates the initiation of dendritic spikes in oDSGCs.

Overall, this is a very interesting study from an excellent group. The technical quality of the experiments is outstanding. Results are clearly illustrated in the figures and explained well in the text. However, the authors need to include recent findings on cholinergic input to DSGCs in the introduction and discussion of their manuscript to put their findings in the appropriate context. In addition, they should further explore and note the influence of stimulus contrast on their results.

Specific comments:

- 1) The authors need to include in the introduction and discussion of their manuscript a recent study by Sethuramanujam et al (Neuron 2016), which revealed the importance of cholinergic transmission from SACs in driving ooDSGC spiking under naturalistic and low contrast stimulus conditions.
- 2) The presentation of the data as cumulative response distributions in Fig4 was counterintuitive to me. The authors should add panels that present response data of oDSGCs and ooDSGCs as a function of stimulus contrast.
- 3) The authors show in Fig5 that the suprathreshold receptive field size shrinks in the presence of cholinergic blockers. The strength of this effect is likely dependent on the stimulus contrast. The authors should explore how the RF size changes as a function of stimulus contrast.
- 4) Given the finding of Sethuramanujam et al that cholinergic input is more dominant at low contrast compared to high contrast the authors should mention the contrast of their stimuli throughout this study.
- 4) A minor point is that I would encourage the authors to replace SBAC with the more widely used abbreviation SAC for starburst amacrine cell.

Reviewer #3 (Remarks to the Author):

The manuscript “Dendro-dendritic cholinergic excitation drives dendritic spike initiation in retinal ganglion cells” by Brombas et al investigates the effect of cholinergic transmission in the starburst amacrine cell to direction selective ganglion cell circuit in the rabbit retina.

The study reports somatic and dendritic patch-clamp recordings of starburst amacrine cells and direction-selective ganglion cells which are an important tool for precisely measuring synaptic signal transmission and details of electrical signal modulation in neurons. The study is technically of high quality and can be of substantial interest. However, I have a major concern that has essentially prevented me from understanding the scope and impact of this study. I was not able to understand how the provided data can be interpreted in the context of the now rather well-established synaptic circuitry in this tissue, namely the number of synaptic contacts between SBACs and DSGCs. This has been precisely established at the electron microscopic level and shown to be dependent on the angle of the SBAC's dendrite and the preferred direction of the DSGC (Briggman et al, 2011). It has been shown that the number of synaptic contacts is about 13 fold higher in the null-direction than in the reversed direction. Based on this circuitry it is not clear how the results presented can be interpreted. In the discussion the authors mention that one may have to consider all the non-synaptic appositions as sites of cholinergic release. However, this then creates a picture of a rather unspecific background cholinergic release in the retina, where the inhibitory signals are the decisive modulators of computationally relevant synaptic transmission. If so, the statements about cholinergic excitation "driving", "powerfully augmenting" etc stimulus sensitivity should be phrased more cautiously. Moreover, when the authors refer to "dendro-dendritic" release, it should be made very clear upfront that this is non-synaptic (if I understand correctly).

As a constructive suggestion I would strongly advise to start with a presentation of the model of the system that is being investigated which builds on the known synaptic circuitry, non-synaptic appositions etc. This would allow the reader to put the presented data into a meaningful context. As it is now, one is left confused.

Secondly, I was not able to extract from the figure legends what exactly is shown in the figures. Clear statements about whether neurons were stimulated using the patch pipettes or using light stimuli are missing, rather the interpretations are presented. Together with the missing reference circuit, this rendered me unable to fully judge the presented data.

Moreover, when stimulated using visual stimuli, somatic recording in a neuron like SBAC (which is expected to have substantial dendritic nonlinear mechanisms, which may not be visible at the soma) it is much less informative.

This lack of precise description has made it virtually impossible to follow the results and determine whether what is shown is an interesting new result or just a speculation based on the notion of dendritic excitability.

Again I would like to emphasize it is well possible that these results are important. They add on to the data existing for the effect of cholinergic modulation for on-off DSGCs. Presenting physiology, however, without precise inclusion of the known circuits is in my view below

standard.

Reviewer #1:

We thank the reviewer for highlighting the relevance of our work, and pinpointing that our work is the first to demonstrate that SAC-mediated cholinergic excitation is locally integrated in the dendritic tree of DSGCs to control dendritic spike generation. In the revised manuscript we have substantially revised the text, and improved the presentation of figures to directly address the reviewers specific comments.

Specific comments (*Reviewers comments in italics*)

1. My major concern is focused on the results shown in Figures 1d-1h. The On DSGCs in these figures are aligned with their preferred directions (upward). Activating SBACs located in the upper side (labeled "Preferred subfield" in Figure 1d) produced a net depolarization in On DSGCs, while activation of most SBACs on the lower side (labeled "Preferred subfield" in Figure 1d) produced a hyperpolarization (Figure 1h). This is directly contradictory to all previous studies on On-Off DSGCs and On DSGCs. It is well-accepted that the strongest inhibitory inputs onto DSGCs come from SBAC dendrites that extend to the null direction of DSGCs. The authors need to address this surprising discrepancy in their manuscript.

We thank the reviewer for highlighting this important point. We have clarified in the text of the revised manuscript the design and interpretation of these data, and have revised **Figure 2** (formally Figure 1 panels d-h) to better illustrate our findings. We believe that our poor data illustration contributed to a misunderstanding. Previous work has highlighted the direction of light movement (see for example ^{1,2}) to delineate SACs located on the preferred- and null-side of the dendritic arbor of ON-OFF-DSGCs. In our analysis we aligned morphologies by the preferred direction vector of action potential output, derived from polar plot analysis (**Figure 2 and Supplementary Figure 2**), which is customary in the field (see for example red lines fit to polar plots in ¹). We, however, failed to adequately highlight that this vector is 180 degrees rotated to the direction of light bar movement. To highlight this we have added schematic light bars indicating the direction of light bar movement to the appropriate Figure (**Figure 2e**), reworked the legend of **Figure 2**, and modified the appropriate text in the Results section (page 5 to 6 of the revised manuscript). We believe that our findings from ON-SAC to ON-DSGCs connections are in line with those previously reported for ON-OFF-DSGCs (page 6 of the revised manuscript). The results section now reads (page 5 to 6):

We next enquired if the synaptic impact of ON-SACs aligned with the directional tuning of postsynaptic ON-DSGCs (**Fig. 2**). To do this we mapped the action potential output of ON-DSGCs generated by light bars moved across their receptive fields and made paired recordings from ON-SACs positioned close to the edge of the dendritic arbor first activated by light stimuli moving in a preferred direction, the preferred dendritic subfield, or ON-SACs positioned close to the edge of the dendritic arbor first activated by null direction light stimuli, the null dendritic subfield (**Fig. 2a,b** inset and **Fig. 2c,d**, inset, **Supplementary Fig. 2**; light bar size= 100 by 300-400 μm , moved in one of 12 directions at 0.24 mm / s; direction selectivity index (DSI)= 0.93 ± 0.03 ; SAC-DSGC soma separation= 270 ± 14 mm; $n= 17$). When the somata of ON-SACs were positioned close to the edge of the dendritic tree of ON-DSGCs first activated by preferred direction light stimuli, and the sites of close dendro-dendritic SAC-DSGC apposition restricted to the preferred dendritic subfield, net excitatory responses were generated (**Fig. 2a-b,e**; PSP amplitude= 2.02 ± 0.37 mV, integral= 31.4 ± 4.3 mV.s; $n= 6$ pairs). In contrast, when the somata and sites of dendro-dendritic apposition were focussed on the null subfield, responses were on average inhibitory, but demonstrated a wide range (**Fig. 2c-e**; PSP amplitude= -0.59 ± 0.8 mV; integral= -33.7 ± 21.0 mV.s; $n= 11$ pairs; significantly different from preferred subfield: $P= 0.035$, $T= 2.31$; integral: $P= 0.041$, $T= 2.24$). Under current-clamp recording conditions, therefore the balance of SAC-mediated cholinergic excitation outweighs that of inhibition in the preferred dendritic subfield of ON-DSGCs, while inhibition outweighs excitation in the null dendritic subfield (**Fig. 2e**). This finding parallels the targeted dendritic impact of SAC-mediated excitation and inhibition reported for ON-OFF DSGCs ¹⁻⁴, and so reveals stereo-typed dendro-dendritic circuitry in the ON- and OFF-sublamina of the inner plexiform layer.

2. The Abstract and Introduction needs to be re-worked to include an updated description of the current literature on this topic. For example, the authors claim that "it remains unknown, however, how SBAC-mediated cholinergic excitation is integrated together with bipolar-cell-mediated glutamatergic excitation and SBAC-mediated inhibition to control the action potential output of DSGCs".

In fact, two studies published earlier this year (Sethuramanujam et al, 2016; Poleg-Polsky and Diamond, 2016) addressed exactly this question, but are omitted in this manuscript. Kittila and Massey, 1995 also needs to be included in the references.

We thank the reviewer for highlighting this important point. We agree that the recent work of Sethuramanujam et al and Poleg-Polsky & Diamond has shed light on the important contribution of cholinergic excitation as a gating mechanism of the voltage-dependent NMDA receptor-mediated conductances - activated by glutamate release from bipolar-cells. We have completely rewritten the Introduction to highlight this point and added careful comment in the Discussion section of our revised manuscript. In the Introductory section of the revised manuscript we indicate (page 3):

SACs are, however, not simply feed-forward inhibitory interneurons, as ultrastructural and functional evidence indicates that both GABA and acetylcholine (ACh) are co-released by SAC to drive postsynaptic inhibition and excitation¹⁻¹³. The physiological role of the co-released neurotransmitter ACh is however less well understood. Previous studies, using somatic recording techniques, have revealed that SAC-mediated excitation does not directly contribute to directional selectivity, but acts non-specifically to enhance the action potential output of DSGCs^{12, 14}. Consistent with this, during development the dendritic subfield-dependent strengthening of dendro-dendritic GABAergic inhibition occurs in parallel with the refinement of the symmetry of cholinergic signalling in DSGCs⁴. SAC-mediated cholinergic signalling has however been demonstrated to powerfully control the action potential output of DSGCs in response to time-varying visual stimuli, and act as an essential excitatory signal under mesopic, dim light, conditions^{15, 16}. Light-evoked feed-forward SAC-mediated cholinergic excitation may therefore function to provide local dendritic depolarization that acts to gate bipolar-cell-mediated glutamatergic excitation of DSGCs¹⁶, which is largely mediated NMDA receptors¹⁷.

Furthermore, in the Discussion section we highlight this work (pages 15-16):

We suggest that SAC-mediated cholinergic excitation acts as a dendritic subunit-specific gate of bipolar-cell signalling, that when activated by moving light bars powerfully controls the action potential output of DSGCs, to define the stimulus sensitivity and receptive field area over which the computation of direction selectivity is executed (**Fig. 10**). An idea supportive of previous observations made from ON-OFF-DSGCs, which have revealed that cholinergic excitation acts to control bipolar-cell mediated glutamatergic excitation by supplying membrane depolarization to relieve the voltage-dependent block of NMDA receptor-mediated PSPs¹⁶. Our findings however extend this idea to demonstrate that these excitatory inputs act in concert to drive the initiation of terminal dendritic spikes, and thus launch a cascade of dendritic spike generation which ultimately results in the driving of action potential output, providing a substrate for the localized dendritic processing of light stimuli¹⁸ (**Fig. 10**).

We apologize for the omission of reference to Kittila & Massey (1997), which has now been appropriately referenced in the revised manuscript.

3. In Discussion, page 12, the third paragraph, the "preferred-side" and "null-side" defined in this paper is opposite to those defined in previous studies. Thus, this paragraph needs to be re-written to address the discrepancy mentioned above.

Please see response to specific comment #1.

4. Page 15, last paragraph: "Support for these electrophysiological data is provided by ultrastructural analysis...." These sentences sound vague to me: what did the authors mean by "the different classes of dendro-dendritic contacts"? Which cell type does "the null-dendritic subtree" refer to? Reference #17 by Briggman and colleagues demonstrates an antiparallel relationship between the orientation of presynaptic SBAC dendrites and the preferred direction of On-Off DSGCs, which is not consistent with the wiring pattern shown in Figure 1 of this paper.

We thank the reviewer for raising this important issue, we have completely revised our discussion of previous ultrastructural work. Careful analysis of the onset time of ON-SAC-evoked cholinergic and GABAergic postsynaptic potentials revealed a significant difference in onset time (revised **Figure 1**), that coupled with previous work^{16, 19} suggests that cholinergic signalling may occur in a local paracrine

fashion in the IPL. We have revised our Discussion section to reflect this new analysis to form a more coherent interpretation of our data (page 17, revised manuscript):

A structural substrate for the dual excitatory and inhibitory roles of SACs in the retinal circuitry is well documented^{1-10, 12, 13}. The dendrites of ON- and OFF-SACs form dense choline acetyl transferase and vesicular ACh transporter delineated fascicles in the inner plexiform layer, the passage of which are closely followed by the dendrites of DSGCs²⁰⁻²². Notably this mapping is most refined for small terminal DSGC dendrites, the predominant site of SAC-DSGC synaptic connectivity²³. Ultrastructural analysis of murine ON-OFF DSGCs has, however, shown that the distribution of dendro-dendritic synapses is asymmetrical, with a ~13-fold greater density established by SACs positioned on the functional determined null dendritic subfield of ON-OFF-DSGCs¹⁹. Our results demonstrate in the simplified circuitry of the ON-sublamina of the inner plexiform layer that the postsynaptic impact of ON-SACs is position dependent, with cholinergic excitation dominating in the preferred, and GABAergic inhibition in the null-dendritic subtree, consistent with previous results from ON-OFF DSGCs^{1-4, 24}. As our paired recordings, and previous results, have revealed the obligatory co-release of ACh and GABA from SACs, these data suggest that ACh may be released from, or have postsynaptic impact at, sites other than dendro-dendritic synapses, a finding that is supported by the relatively long time to onset of cholinergic PSPs. Furthermore, direct dendritic recordings revealed that light stimuli, and direct activation of SACs drove localized nAChR-dependent dendritic excitation, which was integrated locally in the dendritic arbor to control dendritic spike initiation. Our findings are therefore consistent with a model in which ACh operates in a localized paracrine fashion in the IPL¹⁹, at sites constrained by the co-fasciculation of SAC and DSGC dendrites. We therefore conclude that the excitatory and inhibitory neurotransmitters released from the dendrites of feed-forward interneurons in the output layer of the retina are integrated together with direct bipolar-cell-mediated glutamatergic excitation to drive and control the generation of dendritic spikes in DSGCs, the initiation of which determines the light stimulus sensitivity, receptive field size and direction selectivity of action potential output (**Fig. 10**).

Reviewer #2:

We thank the reviewer for their supportive comments indicating that "*Overall, this is a very interesting study from an excellent group. The technical quality of the experiments is outstanding. Results are clearly illustrated in the figures and explained well in the text.*"

In the revised manuscript we have added a large body of new experimental data, substantially revised the text, and improved the presentation of figures to directly address the reviewers specific concerns.

Specific comments (*Reviewers comments in italics*)

1. The authors need to include in the introduction and discussion of their manuscript a recent study by Sethuramanujam et al (Neuron 2016), which revealed the importance of cholinergic transmission from SACs in driving ooDSGC spiking under naturalistic and low contrast stimulus conditions.

We thank the reviewer for highlighting this point we have revised the Introduction and Discussion sections of the manuscript to fully describe the recent Sethuramanujam paper. In the Introductory section of the revised manuscript we indicate (page 3):

SACs are, however, not simply feed-forward inhibitory interneurons, as ultrastructural and functional evidence indicates that both GABA and acetylcholine (ACh) are co-released by SAC to drive postsynaptic inhibition and excitation¹⁻¹³. The physiological role of the co-released neurotransmitter ACh is however less well understood. Previous studies, using somatic recording techniques, have revealed that SAC-mediated excitation does not directly contribute to directional selectivity, but acts non-specifically to enhance the action potential output of DSGCs^{12, 14}. Consistent with this, during development the dendritic subfield-dependent strengthening of dendro-dendritic GABAergic inhibition occurs in parallel with the refinement of the symmetry of cholinergic signalling in DSGCs⁴. SAC-mediated cholinergic signalling has however been demonstrated to powerfully control the action potential output of DSGCs in response to time-varying visual stimuli, and act as an essential excitatory signal under mesopic, low contrast, conditions^{15, 16}. Light-evoked feed-forward SAC-mediated cholinergic excitation may therefore function to provide local dendritic depolarization that acts to gate bipolar-cell-mediated glutamatergic excitation of DSGCs¹⁶, which is largely mediated NMDA receptors¹⁷.

Furthermore, in the Discussion section we highlight this work (pages 15-16):

We suggest that SAC-mediated cholinergic excitation acts as a dendritic subunit-specific gate of bipolar-cell signalling, that when activated by moving light bars powerfully controls the action potential output of DSGCs, to define the stimulus sensitivity and receptive field area over which the computation of direction selectivity is executed (**Fig. 10**). An idea supportive of previous observations made from ON-OFF-DSGCs, which have revealed that cholinergic excitation acts to control bipolar-cell mediated glutamatergic excitation by supplying membrane depolarization to relieve the voltage-dependent block of NMDA receptor-mediated PSPs¹⁶. Our findings however extend this idea to demonstrate that these excitatory inputs act in concert to drive the initiation of terminal dendritic spikes, and thus launch a cascade of dendritic spike generation which ultimately results in the driving of action potential output, providing a substrate for the localized dendritic processing of light stimuli¹⁸ (**Fig. 10**).

2. The presentation of the data as cumulative response distributions in Fig 4 was counterintuitive to me. The authors should add panels that present response data of oDSGCs and ooDSGCs as a function of stimulus contrast.

We have revised the presentation of this material. In the revised manuscript (revised **Figure 5**), moving light bar speed and stimulus intensity relationship are shown for ON-DSGCs (Figure 5 panel a and b). We have omitted data for ON-OFF-DSGCs, in response to reviewer #3 comments. These changes are detailed in the revised Results section (page 8):

To examine if cholinergic excitation influenced the responsiveness of ON-DSGCs to light stimuli across a wide stimulus range, we systematically varied the intensity of light stimuli and their speed of motion to generate stimulus-response relationship under control conditions and in the presence of a

nAChR antagonist (**Fig. 5a-b**; intensity: 10 to 100% above background; speed: 0.04 to 0.9 mm / s). Under control conditions the directional selective output responses of ON-DSGCs emerged at the light stimulus threshold for action potential firing and were maintained across a 50-fold preferred direction firing range (**Fig. 5c**). The antagonism of nAChRs increased the threshold light intensity, constrained the speed of stimuli that generated action potential output, and attenuated action potential firing throughout the range of light stimuli (**Fig. 5a-b**). Notably, this compression of dynamic range was not accompanied by a degradation of the fidelity of the computation of direction selectivity (**Fig. 5c**; DSI: control= 0.946 ± 0.014 ; Hex= 0.990 ± 0.004 ; $n = 70$ trials, $n = 9$ cells). To further explore the light range over which SAC-mediated cholinergic signalling controlled the responsiveness of ON-DSGCs, we made recordings in preparations adapted to mesopic light conditions (background illumination and stimulus intensity reduced to 0.06 of the standard photopic levels). Under these dim light conditions antagonism of nAChRs dramatically reduced the action potential output of ON-DSGCs evoked by preferred and null direction light bars (**Supplementary Fig. 6**; preferred direction: control= 15.1 ± 2.3 Hz; nAChR antagonist= 0.4 ± 0.1 Hz; $P < 0.0001$, $T = 6.314$; $n = 11$; null direction: control= 1.0 ± 0.3 Hz; nAChR antagonist= 0.04 ± 0.02 Hz; $P = 0.0027$, $T = 3.961$). Together these data show that cholinergic excitation acts to control the light stimulus-sensitivity and the magnitude of ON-DSGC output responses.

3. The authors show in Fig5 that the suprathreshold receptive field size shrinks in the presence of cholinergic blockers. The strength of this effect is likely dependent on the stimulus contrast. The authors should explore how the RF size changes as a function of stimulus contrast.

We thank the reviewer for this insight. We have undertaken a major series of experiments to address this concern. In our new data summarized in **Figure 6** we have analysed the impact of cholinergic excitation on the relationship between dendritic field size and supra-threshold receptive field size across a broad range of light stimuli. In addition to the standard photopic 100% stimulus contrast condition reported in the original submission, we have used photopic 50% contrast stimuli, as well as mesopic, dim light stimuli (6% of the intensity of photopic stimuli). As presented in revised Figure 6 and in the Results section (page 9), blockade of nAChRs profoundly disrupted the wide field receptive field properties of ON-DSGCs to a similar degree under photopic (both 100% and 50% stimulus contrast) and mesopic light conditions (compare graphs in **Fig. 6d-e**), suggesting that SAC-mediated cholinergic signalling functions to powerfully drive dendritic excitation across a broad physiological range of stimuli. We have revised our presentation of this material, now showing the displacement between first action potential generation and the edge of the preferred dendritic tree as averaged data and cumulative probability distributions for each trial. In addition we present new Supplementary material detailing the reduction of light responses evoked by mesopic light stimuli (Supplementary Figure 6). These changes are detailed in the revised manuscript (page 9):

To gain insight into how SAC-mediated cholinergic excitation is integrated in postsynaptic ganglion cells we first determined if cholinergic signalling controlled the receptive field structure of ON-DSGCs. Under control conditions we found a tight spatial relationship between the size of the supra-threshold receptive field and the dendritic field size of ON-DSGCs across a wide-range of light conditions, consistent with previous reports²⁵ (**Fig. 6**; receptive field size= $923 \pm 24 \mu\text{m}$; dendritic field size= $851 \pm 23 \mu\text{m}$, $r = 0.574$; $P < 0.0001$, $n = 66$ cells). Notably, when preferred direction light bars were swept across the receptive field a close relationship was found between the time at which light stimuli activated the retinal circuitry which innervated the edge of the preferred dendritic subfield of ON-DSGCs and the time of occurrence of the first light-evoked action potential, which could be converted into a spatial relationship (**Fig. 6a-e**; control: displacement= $19.8 \pm 3.6 \mu\text{m}$; $n = 642$ trials, $n = 66$ cells). The pharmacological blockade of nAChRs reduced action potential firing and delayed its onset as preferred direction light bars were swept across the receptive field, resulting in a disruption of the relationship between dendritic field edge and action potential generation, and a dramatic constriction of supra-threshold receptive field size (**Fig. 6a-e**; control= $923 \pm 24 \mu\text{m}$; nAChR antagonist= $471 \pm 40 \mu\text{m}$; $P < 0.0001$; $T = 11.27$; $n = 66$). Notably the blockade of SAC-mediated cholinergic excitation disrupted the wide field receptive field properties of ON-DSGCs to a similar degree under photopic and mesopic light conditions (compare graphs in **Fig. 6d-e**), suggesting that SAC-mediated cholinergic signalling functions to powerfully drive dendritic excitation across a broad physiological range of stimuli.

4. Given the finding of Sethuramanujam et al that cholinergic input is more dominant at low contrast compared to high contrast the authors should mention the contrast of their stimuli throughout this study.

We thank the reviewer for highlighting this point, we have indicated stimulus intensity relative to background, as well as the adaptation state of the retina, throughout the revised manuscript.

5. A minor point is that I would encourage the authors to replace SBAC with the more widely used abbreviation SAC for starburst amacrine cell.

We have used the abbreviation SAC for Starburst amacrine cell throughout the revised manuscript.

Reviewer #3:

We thank the reviewer for indicating that "*The study is technically of high quality and can be of substantial interest*".

We apologize to the reviewer for the poor description of our work in the original submission. We have undertaken a body of additional experimental work, and have rewritten large sections of the manuscript and clarified our illustrated materials to directly address the reviewers concerns, stated as "*However, I have a major concern that has essentially prevented me from understanding the scope and impact of this study*".

Specific comments (*Reviewers comments in italics*)

1. I was not able to understand how the provided data can be interpreted in the context of the now rather well-established synaptic circuitry in this tissue, namely the number of synaptic contacts between SBACs and DSGCs. This has been precisely established at the electron microscopic level and shown to be dependent on the angle of the SBAC's dendrite and the preferred direction of the DSGC (Briggman et al, 2011). It has been shown that the number of synaptic contacts is about 13 fold higher in the null-direction than in the reversed direction. Based on this circuitry it is not clear how the results presented can be interpreted. In the discussion the authors mention that one may have to consider all the non-synaptic appositions as sites of cholinergic release. However, this then creates a picture of a rather unspecific background cholinergic release in the retina, where the inhibitory signals are the decisive modulators of computationally relevant synaptic transmission. If so, the statements about cholinergic excitation "driving", "powerfully augmenting" etc stimulus sensitivity should be phrased more cautiously. Moreover, when the authors refer to "dendro-dendritic" release, it should be made very clear upfront that this is non-synaptic (if I understand correctly).

We thank the reviewer for highlighting these important issues. We have thoroughly revised the manuscript to directly address these concerns.

First we have rewritten the Introduction section of the manuscript to better describe the known SAC-DSGC circuitry in the inner plexiform layer of the retina, highlighting previous ultrastructural work. These issues are summarised on pages 2 to 3 of the Introduction:

The direction-selective action potential output of DSGCs is believed to be computed by the integration of a directionally un-tuned excitatory input, predominately mediated by glutamate release from bipolar cells²⁶⁻²⁸, with a directionally tuned inhibitory synaptic input generated by the dendritic release of GABA from axonless feed-forward interneurons, termed starburst amacrine cells (SACs)^{3, 19, 29-31}. SACs represent essential components of the direction-selective circuitry of the retina³². Both ON- and OFF-SACs are distributed, in a mosaic throughout the retina³³, and possess a unique radial dendritic morphology, in which each dendrite operates in electrical isolation^{29, 34, 35}. Functionally, two-photon calcium imaging has revealed that SAC dendritic calcium responses are preferentially generated when light stimuli move from the soma toward terminal dendritic sites²⁹, a direction-selective calcium signal that is thought to generate the dendritic release of neurotransmitters from terminal dendritic synaptic output zones^{12, 23, 36}. The directional tuning of DSGCs is disrupted by the pharmacological antagonism of GABA_A receptors^{11, 12}, evincing a prominent role of SAC-mediated synaptic inhibition. Consistent with this, electrophysiological and high-resolution morphological studies have demonstrated that the SAC-mediated inhibitory synaptic control of null direction light responses is mediated by a greater GABA_A receptor-mediated synaptic conductance, synapse number, and distribution of dendro-dendritic synapses in the null dendritic subfield of DSGCs^{1, 3, 4, 19, 24}. SACs are, however, not simply feed-forward inhibitory interneurons, as ultrastructural and functional evidence indicates that both GABA and acetylcholine (ACh) are co-released by SAC to drive postsynaptic inhibition and excitation¹⁻¹³. The physiological role of the co-released neurotransmitter ACh is however less well understood. Previous studies, using somatic recording techniques, have revealed that SAC-mediated excitation does not directly contribute to directional selectivity, but acts non-specifically to enhance the action potential output of DSGCs^{12, 14}. Consistent with this, during development the dendritic subfield-dependent strengthening of dendro-dendritic GABAergic inhibition occurs in parallel with the refinement of the symmetry of cholinergic signalling in DSGCs⁴. SAC-mediated cholinergic signalling has however been demonstrated to powerfully control the action potential output of DSGCs in response to time-varying visual stimuli, and act as an essential excitatory signal under mesopic, low contrast,

conditions^{15, 16}. Light-evoked feed-forward SAC-mediated cholinergic excitation may therefore function to provide local dendritic depolarization that acts to gate bipolar-cell-mediated glutamatergic excitation of DSGCs¹⁶, which is largely mediated NMDA receptors¹⁷.

Second, we have undertaken detailed analysis of the time course of ON-SAC evoked excitatory and inhibitory postsynaptic potentials, revealing that the time to onset of ON-SAC evoked cholinergic EPSPs is longer than those of GABAergic IPSPs. We interpret these findings in line with previous literature to indicate that cholinergic signalling occurs in a localised paracrine fashion in the inner plexiform layer. We have specifically addressed this point in **Figure 1** of the revised manuscript, and describe these results in the Results section (pages 4-5):

Pharmacologically isolated cholinergic excitatory and GABAergic inhibitory PSPs exhibited a characteristic difference in their time to onset following SAC activation, with the onset of GABAergic inhibitory PSPs leading that of cholinergic excitatory PSPs (**Fig. 1f-h, Supplementary Fig. 1a-b**; IPSP onset time= 6.6 ± 0.9 ms; EPSP onset time= 10.3 ± 1.3 ms, $P = 0.0432$; $T = 2.295$; onset time measured at 5% of PSP amplitude; IPSP rise time= 8.9 ± 1.9 ms; EPSP rise time= 5.8 ± 0.5 ms; rise time measured between 10-90% of PSP amplitude; $n = 14$). This difference could not be accounted for by a presynaptic mechanism, such as temporal jitter in the engagement of dendritic transmitter release, as the distribution of onset latencies around the mean was similar for pharmacologically isolated excitatory and inhibitory PSPs (**Supplementary Fig. 1c**). The onset time of SAC-evoked PSPs could however be influenced by the time course of transmitter diffusion from SAC dendritic sites. Consistent with this previous studies have suggested cholinergic signalling may occur in a paracrine fashion in the retina, and so may not be reliant on structural determined dendro-dendritic synapses^{16, 19}, an idea consistent with the observed slower onset time of cholinergic excitation. Functionally, the different onset of dendro-dendritic excitation and inhibition would be predicted to lead to the generation of biphasic compound PSPs, if both GABA and ACh are co-released from a single ON-SAC. Indeed when pharmacologically isolated excitatory and inhibitory components were arithmetically summed, a clear biphasic compound PSP was generated (**Fig. 1h**). Consistent with this, SAC-evoked PSPs recorded under control conditions frequently exhibited a biphasic waveform, that was clearly evident in single trials and when consecutive responses were digitally averaged (**Fig. 1d & h**, respectively). Together, these data reveal that ACh and GABA are co-released from single ON-SACs.

Third, we have rewritten the Discussion section of the revised manuscript to indicate how our new data can be integrated together with previous electrophysiological and structural findings. The close of the Discussions section now reads (page 17):

A structural substrate for the dual excitatory and inhibitory roles of SACs in the retinal circuitry is well documented^{1-10, 12, 13}. The dendrites of ON- and OFF-SACs form dense choline acetyl transferase and vesicular ACh transporter delineated fascicles in the inner plexiform layer, the passage of which are closely followed by the dendrites of DSGCs²⁰⁻²². Notably this mapping is most refined for small terminal DSGC dendrites, the predominant site of SAC-DSGC synaptic connectivity²³. Ultrastructural analysis of murine ON-OFF DSGCs has, however, shown that the distribution of dendro-dendritic synapses is asymmetrical, with a ~13-fold greater density established by SACs positioned on the functional determined null dendritic subfield of ON-OFF-DSGCs¹⁹. Our results demonstrate in the simplified circuitry of the ON-sublamina of the inner plexiform layer that the postsynaptic impact of ON-SACs is position dependent, with cholinergic excitation dominating in the preferred, and GABAergic inhibition in the null-dendritic subtree, consistent with previous results from ON-OFF-DSGCs^{1-4, 24}. As our paired recordings, and previous results, have revealed the obligatory co-release of ACh and GABA from SACs, these data suggest that ACh may be released from, or have postsynaptic impact at, sites other than dendro-dendritic synapses, a finding that is supported by the relatively long time to onset of cholinergic PSPs. Furthermore, direct dendritic recordings revealed that light stimuli, and direct activation of SACs drove localized nAChR-dependent dendritic excitation, which was integrated locally in the dendritic arbor to control dendritic spike initiation. Our findings are therefore consistent with a model in which ACh operates in a localized paracrine fashion in the IPL¹⁹, at sites constrained by the co-fasciculation of SAC and DSGC dendrites. We therefore conclude that the excitatory and inhibitory neurotransmitters released from the dendrites of feed-forward interneurons in the output layer of the retina are integrated together with direct bipolar-cell-mediated glutamatergic excitation to drive and control the generation of dendritic spikes in DSGCs, the initiation of which determines the light stimulus sensitivity, receptive field size and direction selectivity of action potential output (**Fig. 10**).

2. *As a constructive suggestion I would strongly advise to start with a presentation of the model of the system that is being investigated which builds on the known synaptic circuitry, non-synaptic appositions etc. This would allow the reader to put the presented data into a meaningful context. As it is now, one is left confused.*

We thank the reviewer for this suggestion, we have included a cartoon of the basic retinal circuitry activated by light stimuli in the revised manuscript (**Figure 3a**, inset). We have produced a schematic figure that summarizes our findings and interpretations, which we use to guide the reader through the Discussion section of the manuscript (**Figure 10**). Furthermore, we have modified the final paragraph of the Discussion section as noted in response to specific comment #1.

3. *I was not able to extract from the figure legends what exactly is shown in the figures. Clear statements about whether neurons were stimulated using the patch pipettes or using light stimuli are missing, rather the interpretations are presented. Together with the missing reference circuit, this rendered me unable to fully judge the presented data.*

We have revised the presentation of the Figures throughout the revised manuscript, and rewritten the figure legends to more clearly indicate the illustrated material.

4. *Moreover, when stimulated using visual stimuli, somatic recording in a neuron like SBAC (which is expected to have substantial dendritic nonlinear mechanisms, which may not be visible at the soma) it is much less informative.*

We apologize to the reviewer for the lack of clarity in our original submission. All recordings from SACs were made from the somata. We did not directly investigate dendritic mechanisms of SACs. All presented simultaneous somato-dendritic recordings were made from ON-DSGCs. In some cases, this was complemented by presynaptic somatic recordings from SACs to demonstrate that ACh release from ON-SACs evokes localized dendritic EPSPs, which were capable of directly driving dendritic spike generation in postsynaptic ON-DSGCs. We have rewritten the Results section to clarify this issue (pages 16 to 17):

To explore the determinants of this form of sub-cellular integration, we tested if cholinergic excitation provided by a single SAC was capable of driving the initiation of dendritic spikes in ON-DSGCs. To do this we made simultaneous somato-dendritic recordings from ON-DSGCs and a third somatic recording from a presynaptic ON-SAC, in the presence of the GABA_A receptor antagonist GABAZINE (**Fig. 7e-g**; $n=7$). Under these conditions the activation of a single SAC led to the generation of nAChR-mediated dendritic excitatory PSPs, which were crowned by the firing of dendritic spikes, in recordings where dendro-dendritic SAC-DSGC appositions were focussed in the recorded dendritic arbor of the ON-DSGCs at loci distal to the site of dendritic recording (**Fig. 7e-g**; dendritic recordings 145 ± 5 mm from soma; average site of SAC dendro-dendritic appositions from DSGC soma = 437 ± 26 mm; $n=5$). Simultaneous somatic recording revealed that each dendritic spike preceded and drove action potential firing (dendritic spike to action potential delay = 0.40 ± 0.05 ms). In contrast, when dendritic recordings were made from the dendritic subfield contralateral to the site of predominate SAC innervation, SAC-evoked cholinergic excitation drove action potential firing, which was first recorded somatically and subsequently back-propagated to the dendritic recording site, consistent with cholinergic excitation of the contralateral dendritic tree (**Supplementary Fig. 8**). Taken together these data directly demonstrate that dendro-dendritic cholinergic excitation is capable of driving the generation of dendritic spikes in ON-DSGCs.

5. *This lack of precise description has made it virtually impossible to follow the results and determine whether what is shown is an interesting new result or just a speculation based on the notion of dendritic excitability. Again I would like to emphasize it is well possible that these results are important. They add on to the data existing for the effect of cholinergic modulation for on-off DSGCs. Presenting physiology, however, without precise inclusion of the known circuits is in my view below standard.*

We thank the reviewer for this important comment. We have thoroughly revised the Introductory, Results and Discussion sections of the revised manuscript and carefully revised the main and supplementary figures to carefully explain and illustrate our findings. We hope that the new presentation, and Discussion of the local paracrine release of ACh helps to bridge the gap between our

physiological findings and previous high-resolution circuit mapping experiments. Indeed, previous high-resolution circuit mapping studies have found it difficult to explain the role of SAC-mediated cholinergic signalling in the output layer of the retina, see discussion in ¹⁹. We hope that our new analysis and more thorough discussion better explains the role of SAC-mediated cholinergic signalling in the retina, by demonstrating how local integration within the dendritic tree of ON-DSGCs controls this class of RGC action potential output. To avoid confusion between our study and previous work, we have removed all functional data concerning the role of cholinergic signalling in controlling the output properties of rabbit ON-OFF-DSGCs in the revised manuscript. We hope that our revised presentation and discussion of material, which is summarised in a new schematic diagram (**Figure 10**), place our new findings into a logical and understandable framework.

References

1. Wei, W., Hamby, A.M., Zhou, K. & Feller, M.B. Development of asymmetric inhibition underlying direction selectivity in the retina. *Nature* **469**, 402-406 (2011).
2. Pei, Z., et al. Conditional knock-out of vesicular GABA transporter gene from starburst amacrine cells reveals the contributions of multiple synaptic mechanisms underlying direction selectivity in the retina. *J Neurosci* **35**, 13219-13232 (2015).
3. Lee, S., Kim, K. & Zhou, Z.J. Role of ACh-GABA cotransmission in detecting image motion and motion direction. *Neuron* **68**, 1159-1172 (2010).
4. Yonehara, K., et al. Spatially asymmetric reorganization of inhibition establishes a motion-sensitive circuit. *Nature* **469**, 407-410 (2011).
5. Ariel, M. & Daw, N.W. Pharmacological analysis of directionally sensitive rabbit retinal ganglion cells. *J Physiol* **324**, 161-185 (1982).
6. Kittila, C.A. & Massey, S.C. Pharmacology of directionally selective ganglion cells in the rabbit retina. *J Neurophysiol* **77**, 675-689 (1997).
7. Brecha, N., Johnson, D., Peichl, L. & Wassle, H. Cholinergic amacrine cells of the rabbit retina contain glutamate decarboxylase and gamma-aminobutyrate immunoreactivity. *Proc Natl Acad Sci USA* **85**, 6187-6191 (1988).
8. Vaney, D.I. & Young, H.M. GABA-like immunoreactivity in NADPH-diaphorase amacrine cells of the rabbit retina. *Brain Res* **474**, 380-385 (1988).
9. O'Malley, D.M. & Masland, R.H. Co-release of acetylcholine and gamma-aminobutyric acid by a retinal neuron. *Proc Natl Acad Sci USA* **86**, 3414-3418 (1989).
10. Feller, M.B., Wellis, D.P., Stellwagen, D., Werblin, F.S. & Shatz, C.J. Requirement for cholinergic synaptic transmission in the propagation of spontaneous retinal waves. *Science* **272**, 1182-1187 (1996).
11. Massey, S.C., Linn, D.M., Kittila, C.A. & Mirza, W. Contributions of GABAA receptors and GABAC receptors to acetylcholine release and directional selectivity in the rabbit retina. *Vis Neurosci* **14**, 939-948 (1997).
12. He, S. & Masland, R.H. Retinal direction selectivity after targeted laser ablation of starburst amacrine cells. *Nature* **389**, 378-382 (1997).
13. Fried, S.I., Munch, T.A. & Werblin, F.S. Directional selectivity is formed at multiple levels by laterally offset inhibition in the rabbit retina. *Neuron* **46**, 117-127 (2005).
14. Chiao, C.C. & Masland, R.H. Starburst cells nondirectionally facilitate the responses of direction-selective retinal ganglion cells. *J Neurosci* **22**, 10509-10513 (2002).
15. Grzywacz, N.M., Amthor, F.R. & Merwine, D.K. Necessity of acetylcholine for retinal directionally selective responses to drifting gratings in rabbit. *J Physiol* **512**, 575-581 (1998).
16. Sethuramanujam, S., et al. A central role for mixed acetylcholine/GABA transmission in direction coding in the retina. *Neuron* **90**, 1243-1256 (2016).
17. Poleg-Polsky, A. & Diamond, J.S. NMDA receptors multiplicatively scale visual signals and enhance directional motion discrimination in retinal ganglion cells. *Neuron* **89**, 1277-1290 (2016).
18. Barlow, H.B. & Levick, W.R. The mechanism of directionally selective units in rabbit's retina. *J Physiol* **178**, 477-504 (1965).
19. Briggman, K.L., Helmstaedter, M. & Denk, W. Wiring specificity in the direction-selectivity circuit of the retina. *Nature* **471**, 183-188 (2011).
20. Brandon, C. Cholinergic neurons in the rabbit retina: dendritic branching and ultrastructural connectivity. *Brain Res* **426**, 119-130. (1987).
21. Vaney, D.I. & Pow, D.V. The dendritic architecture of the cholinergic plexus in the rabbit retina: selective labeling by glycine accumulation in the presence of sarcosine. *J Comp Neurol* **421**, 1-13 (2000).
22. Dong, W., Sun, W., Zhang, Y., Chen, X. & He, S. Dendritic relationship between starburst amacrine cells and direction-selective ganglion cells in the rabbit retina. *J Physiol* **556**, 11-17 (2004).
23. Famiglietti, E.V. Synaptic organization of starburst amacrine cells in rabbit retina: analysis of serial thin sections by electron microscopy and graphic reconstruction. *J Comp Neurol* **309**, 40-70. (1991).
24. Morrie, R.D. & Feller, M.B. An asymmetric increase in inhibitory synapse number underlies the development of a direction selective circuit in the retina. *J Neurosci* **35**, 9281-9286 (2015).
25. Yang, G. & Masland, R.H. Receptive fields and dendritic structure of directionally selective retinal ganglion cells. *J Neurosci* **14**, 5267-5280. (1994).

26. Chen, M., Lee, S., Park, S.J., Looger, L.L. & Zhou, Z.J. Receptive field properties of bipolar cell axon terminals in direction-selective sublaminae of the mouse retina. *J Neurophysiol* **112**, 1950-1962 (2014).
27. Park, S.J., Kim, I.J., Looger, L.L., Demb, J.B. & Borghuis, B.G. Excitatory synaptic inputs to mouse on-off direction-selective retinal ganglion cells lack direction tuning. *J Neurosci* **34**, 3976-3981 (2014).
28. Yonehara, K., *et al.* The first stage of cardinal direction selectivity is localized to the dendrites of retinal ganglion cells. *Neuron* **79**, 1078-1085 (2013).
29. Euler, T., Detwiler, P.B. & Denk, W. Directionally selective calcium signals in dendrites of starburst amacrine cells. *Nature* **418**, 845-852 (2002).
30. Fried, S.I., Munch, T.A. & Werblin, F.S. Mechanisms and circuitry underlying directional selectivity in the retina. *Nature* **420**, 411-414 (2002).
31. Sun, L.O., *et al.* Functional assembly of accessory optic system circuitry critical for compensatory eye movements. *Neuron* **86**, 971-984 (2015).
32. Yoshida, K., *et al.* A key role of starburst amacrine cells in originating retinal directional selectivity and optokinetic eye movement. *Neuron* **30**, 771-780 (2001).
33. Lefebvre, J.L., Kostadinov, D., Chen, W.V., Maniatis, T. & Sanes, J.R. Protocadherins mediate dendritic self-avoidance in the mammalian nervous system. *Nature* **488**, 517-521 (2012).
34. Miller, R.F. & Bloomfield, S.A. Electroanatomy of a unique amacrine cell in the rabbit retina. *Proc Natl Acad Sci U S A* **80**, 3069-3073 (1983).
35. Tukker, J.J., Taylor, W.R. & Smith, R.G. Direction selectivity in a model of the starburst amacrine cell. *Vis Neurosci* **21**, 611-625 (2004).
36. Vlasits, A.L., *et al.* A Role for Synaptic Input Distribution in a Dendritic Computation of Motion Direction in the Retina. *Neuron* **89**, 1317-1330 (2016).

Reviewers' Comments:

Reviewer #1 (Remarks to the Author):

This manuscript highlights the importance of cholinergic inputs from SACs in initiating local dendritic spikes in DSGCs. The combination of dendritic and somatic recordings from DSGCs and stimulation of SACs is technically elegant and impressive, and provides direct evidence of ACh-dependent dendritic spike generation in DSGCs. The authors have added new experiments and changed the text to address the points I have raised. However, I still have several concerns, mainly about the conceptual framework related to this study, and interpretation of the data.

1. The authors describe the SAC-DSGC connectivity pattern in terms of "preferred or null dendritic subfield of DSGCs". This is not consistent with the current models of direction selectivity. In the current models, at least for On-Off DSGCs, the asymmetric wiring between SACs and DSGCs arises from dendritic branches of SACs, not those of DSGCs. Indeed, the dendritic field of a DSGC is considered rather uniform along the preferred-null axis, and the entire dendritic tree receives inhibitory inputs from SAC dendrites oriented in the null direction of the DSGC. Therefore, the use of "preferred or null dendritic subfield of DSGCs" throughout the manuscript (e.g. in pages 2, 5, 14) is inappropriate.

2. Page 2, second paragraph first sentence, the statement about the predominant excitatory input being glutamatergic is not accurate. The importance of cholinergic excitation has been highlighted in numerous published studies (e.g. Sethuramanujam et al, 2016, Weng et al, 2005, Grzywacz et al 1998, Kittila and Massey etc).

3. Page 3 line 6-8, "... SAC-mediated excitation does not directly contribute to direction selectivity, but acts non-specifically to enhance the action potential output of DSGCs." This is not definitively proven. Instead, a role of SAC-mediated cholinergic excitation in direction selectivity has been clearly demonstrated in Grzywacz et al 1998, Lee et al 2010 and Pei et al 2015.

4. The difference in the time to onset for GABAergic and cholinergic PSPs is interesting, and the authors use this to argue for the paracrine nature of cholinergic signaling. However, their result directly contradicts the findings by Lee et al 2010, who demonstrated that the cholinergic EPSCs and GABAergic IPSCs in DSGCs during paired SAC-DSGC recording exhibit the same latency of onset. Lee et al also convincingly demonstrated monosynaptic, calcium-dependent fast synaptic transmission between SACs and DSGCs using ACh. In contrast, no evidence of paracrine SAC-DSGC signaling has been shown in the literature. The authors should substantiate their hypothesis with more evidence, and perform voltage clamp recordings in SAC-On DSGC

pairs to address this discrepancy.

5. Page 6 last line, it is unclear to me how "stereo-typed dendro-dendritic circuitry in the On- and Off-sublamina of the inner plexiform layer" is supported, because the asymmetric wiring between Off-SACs and DSGCs has not been directly demonstrated.

6. Fig. 5c, the authors showed that Hex reduces direction index, a commonly used measure of directional tuning in DSGCs, and thus concluded in the text that Hex does not affect "the fidelity of the computation of direction selectivity". However, I think a more careful interpretation of direction index is necessary. For example, if there is zero or very few spikes in the null direction in the control condition, the null direction firing cannot be reduced further by Hex, while the preferred direction firing is greatly reduced (as shown in Fig. 5c). In this scenario, the direction index is not changed, but the difference between null and preferred direction firing rates is dramatically altered, which could indeed be interpreted as a reduction in the fidelity of the computation of direction selectivity.

Minor points:

7. What do individual data points represent in Fig 1c and 1e?

8. Fig. 2b and d, the polar plots should be rotated 180 degrees so that the vector sums of the spiking responses point to the same preferred direction shown in Fig 2a and 2c.

Reviewer #2 (Remarks to the Author):

The authors have satisfactorily addressed my previous comments. Congratulations on a very nice study

Reviewer #3 (Remarks to the Author):

The authors have taken the concerns of the other reviewers and myself very seriously and have very successfully revised the manuscript. The description of the circuit context and the significance of the authors' findings has much improved. Also the figures are now clearly understandable, allowing a proper assessment of the presented data. As such this is a very successful revision and I can recommend publication without hesitation.

We thank the Reviewers of our manuscript for their comments. We note that Reviewer #2 and Reviewer #3 were entirely satisfied with our revised manuscript, indicating:

Reviewer #2: "The authors have satisfactorily addressed my previous comments. Congratulations on a very nice study."

Reviewer #3: "The authors have taken the concerns of the other reviewers and myself very seriously and have very successfully revised the manuscript. The description of the circuit context and the significance of the authors' findings has much improved. Also the figures are now clearly understandable, allowing a proper assessment of the presented data. As such this is a very successful revision and I can recommend publication without hesitation."

We thank Reviewer #1 for their further interest in our work. Please find below our point-by-point reply.

Reviewer #1 Major concerns:

1. The authors describe the SAC-DSGC connectivity pattern in terms of "preferred or null dendritic subfield of DSGCs". This is not consistent with the current models of direction selectivity. In the current models, at least for On-Off DSGCs, the asymmetric wiring between SACs and DSGCs arises from dendritic branches of SACs, not those of DSGCs. Indeed, the dendritic field of a DSGC is considered rather uniform along the preferred-null axis, and the entire dendritic tree receives inhibitory inputs from SAC dendrites oriented in the null direction of the DSGC. Therefore, the use of "preferred or null dendritic subfield of DSGCs" throughout the manuscript (e.g. in pages 2, 5, 14) is inappropriate.

Previous work detailing the impact of SAC-mediated synaptic input to ON-OFF DSGCs have differentiated the preferred and null-sides of the dendritic arbor of ON-OFF DSGCs using an approach similar to the one illustrated and described in our manuscript¹⁻⁴. We therefore consider the approach we have adopted to describe the dendritic field of ON-DSGCs to be standard in the field, and as so will allow a direct comparison of our work with that previously reported for ON-OFF DSGCs¹⁻⁴.

2. Page 2, second paragraph first sentence, the statement about the predominant excitatory input being glutamatergic is not accurate. The importance of cholinergic excitation has been highlighted in numerous published studies (e.g. Sethuramanujam et al, 2016, Weng et al, 2005, Grzywacz et al 1998, Kittila and Massey etc).

and

3. Page 3 line 6-8, "... SAC-mediated excitation does not directly contribute to direction selectivity, but acts non-specifically to enhance the action potential output of DSGCs." This is not definitively proven. Instead, a role of SAC-mediated cholinergic excitation in direction selectivity has been clearly demonstrated in Grzywacz et al 1998, Lee et al 2010 and Pei et al 2015.

We thank the reviewer for highlight the involvement of the SAC-mediated cholinergic signalling in the direction selective circuitry of the retina. We acknowledge that previous studies have demonstrated a variable contribution of cholinergic signalling to the light-evoked excitation of DSGCs. We also acknowledge that there exists controversy concerning the role of cholinergic signalling in the computation of direction selectivity. As this body of literature motivated our study, we have simplified the Introductory section of the revised manuscript to state as clearly, and concisely as possible existing work (page 3):

The physiological role of the co-released neurotransmitter ACh is however less well understood, and controversy remains on the contribution of this feed-forward excitatory signal to the generation of light-evoked DSGC action potential output, and its role in the computation of direction selectivity^{1, 2, 4-13}.

4. The difference in the time to onset for GABAergic and cholinergic PSPs is interesting, and the authors use this to argue for the paracrine nature of cholinergic signaling. However, their result directly contradicts the findings by Lee et al 2010, who demonstrated that the cholinergic EPSCs and GABAergic IPSCs in DSGCs during paired SAC-DSGC recording exhibit the same latency of onset. Lee et al also convincingly demonstrated monosynaptic, calcium-dependent fast synaptic transmission between SACs and DSGCs using ACh. In contrast, no evidence of paracrine SAC-DSGC signaling has been shown in the literature. The authors should substantiate their hypothesis with more evidence, and perform voltage clamp recordings in SAC-On DSGC pairs to address this discrepancy.

Latency Analysis: Our paired ON-SAC to ON-DSGC recordings revealed that activation of a single SAC evoked nAChR-mediated EPSPs and GABA_A receptor-mediated IPSPs in postsynaptic ON-DSGCs. In extension of the results of Lee et al. 2010¹ we demonstrate that the both cholinergic and GABAergic **synaptic potentials** exhibit fast rise and decay kinetics. (Figs. 1 and 2). Our analysis of the onset latency of cholinergic **EPSPs** and GABAergic **IPSPs** recorded under current-clamp conditions was achieved using **pharmacological separation**. These findings were however confirmed by the appearance of SAC-evoked biphasic synaptic potentials under control conditions (Figure 1 panel h). Our approach contrasts with that used by Lee et al. 2010 who: *i) employed whole-cell somatic voltage clamp techniques and ii) did not pharmacologically separate SAC-evoked excitatory and inhibitory synaptic currents.*

We believe that these two points are of the upmost significance as we have previously experimentally demonstrated that somatic voltage-clamp techniques profoundly distorts the amplitude, reversal potential, kinetics and ability to separate temporally overlapping excitatory and inhibitory of dendritically generated synaptic input in central neurons (Williams and Mitchell, Nature Neurosci. 2008)¹⁴. Such direct experimental analysis of the distortions imposed by somatic voltage-clamp recording techniques have been confirmed in DSGCs by computer simulations (Poleg-Polsky & Diamond, Plos One, 2011)¹⁵.

We note that both these studies have demonstrated that the somatic voltage clamp does not universally control voltage in the dendritic tree, allowing escape potentials to be generated at the dendritic site of synaptic activation. This limitation of the somatic

voltage clamp recording technique has a profound impact when both excitatory and inhibitory synaptic inputs are co-generated^{14, 15} - as is the case in the recordings of Lee et al. 2010¹. We note that as Lee et al. 2010¹ employed somatic voltage clamp techniques, and used the control of the somatic holding potential to "isolate" SAC-evoked excitatory and inhibitory synaptic currents, considerable errors must be considered in their latency analysis, as at the dendritic site of synaptic activation the local membrane potential is not voltage controlled and so free to be charged by excitatory and inhibitory synaptic input, **an effect that will distort latency measurement of "isolated" components**. We therefore respectfully indicate: i) previous measurement of the latency of temporally overlapping SAC-evoked IPSCs and EPSCs recorded by Lee et al. 2010¹ are subject to considerable uncertainty, and ii) somatic voltage-clamp techniques do not represent an adequate tool to address this problem.

We contend that our findings illustrated in Figure 1 and Supplementary Figure 1 using current clamp recordings of pharmacologically isolated SAC-evoked EPSPs and IPSPs represents the most accurate method available to undertake latency analysis. We also note that we have demonstrated that a clear separation of latency is stable across many trials (Figure 1 panel f) and cannot be accounted for by other processes (Supplementary Fig. 1). Furthermore, such latency analysis yielded a prediction concerning the waveform of PSPs evoked under control conditions. Analysis of control SAC-evoked PSPs supported this prediction (Figure 1, panel h).

Evidence for paracrine ACh signalling in the literature: The reviewer indicates that: "In contrast, no evidence of paracrine SAC-DSGC signaling has been shown in the literature."

We contend that there is substantive evidence for a local-paracrine action of ACh in the direction selective circuitry of the retina, to which our direct observations add. We note that discussion of this issue has been addressed in the most recent paper to explore SAC function in the retina, published in late 2016 in the journal *Neuron*¹³, a reference that the reviewer cites at other points in their review.

We note that these authors indicate that SAC signalling in the preferred subfield of ON-OFF-DSGCs to be paracrine (see Fig. 1B of¹³). We further note that the Discussion section of this paper indicates (reproduced from¹³):

"As suggested in previous studies (Briggman et al., 2011), isotropic excitation could be an outcome of the paracrine nature of ACh transmission. The dense plexus of SAC dendrites releasing ACh (with each point containing overlapping dendrites originating from 30–60 SACs; Keeley et al., 2007) and the diffuse expression of acetylcholinesterase (Nichols and Koelle, 1968) together promote paracrine transmission of ACh in the retina (Ariel and Daw, 1982; Ford et al., 2012; Schmidt et al., 1987). Paracrine transmission would allow DSGCs to pool cholinergic signals arising from many dendrites orientated in different directions. In contrast to ACh, the clearance of GABA from the synaptic cleft relies on strong uptake mechanisms that confine its action to the synapse. In this way, co-release of ACh and GABA by SACs could lead to distinct spatiotemporal patterns of activity at the level of the DSGC." Reproduced from¹³.

Indeed Ford et al. 2012 has provided direct evidential support for a role of local paracrine SAC-mediated cholinergic signalling in the developing retina ¹⁶. When taken together with our new findings, which provide direct evidence in support of a local-paracrine action of ACh, we with respect strongly disagree with the reviewers point that "*..no evidence of paracrine SAC-DSGC signaling has been shown in the literature.*" But concur with Sethuramanujam et al. 2016 that new tools are required to directly address this issue, we have therefore stated in the revised discussion section of the manuscript (page 25):

Our findings are therefore consistent with a model in which ACh operates in a localized paracrine fashion in the IPL ^{13, 17 16}, at sites constrained by the co-fasciculation of SAC and DSGC dendrites. The development of new tools for the direct visualization of ACh release and diffusion are, however, required to definitively address this issue.

We further note that our latency analysis, and detailed description of the structure of the IPL was called for by Reviewer # 3 in the first round of review, who indicated that they were entirely happy with the revised manuscript.

5. Page 6 last line, it is unclear to me how "stereo-typed dendro-dendritic circuitry in the On- and Off-sublamina of the inner plexiform layer" is supported, because the asymmetric wiring between Off-SACs and DSGCs has not been directly demonstrated.

Previous work has highlighted the direction-selectivity of OFF responses in DSGCs, and ultra structural data has highlighted asymmetric wiring of OFF-SACs (see Figure 4 of Briggman et al. 2011) ¹⁷.

6. Fig. 5c, the authors showed that Hex reduces direction index, a commonly used measure of directional tuning in DSGCs, and thus concluded in the text that Hex does not affect "the fidelity of the computation of direction selectivity". However, I think a more careful interpretation of direction index is necessary. For example, if there is zero or very few spikes in the null direction in the control condition, the null direction firing cannot be reduced further by Hex, while the preferred direction firing is greatly reduced (as shown in Fig. 5c). In this scenario, the direction index is not changed, but the difference between null and preferred direction firing rates is dramatically altered, which could indeed be interpreted as a reduction in the fidelity of the computation of direction selectivity.

We thank the reviewer for highlighting this issue. We entirely agree that it is necessary to document both the control of the firing rate and the computation of direction selectivity. **That is why our results contains analysis of both parameters, which are illustrated together graphically (Fig. 5, panel c).** We note that the calculation of direction selective indices is standard in the field, but also note that we are the first to produce a coherent analytical and graphical representation of direction selective indices and firing rate to illustrate the role of cholinergic signalling across a wide-range of light stimuli.

Minor points:

7. What do individual data points represent in Fig 1c and 1e?

The indicated cumulative probability distributions represent analysis of the amplitude of each SAC-evoked synaptic potential recorded under the indicated conditions, the indicated number of paired recordings is documented. We note the synaptic potentials that reach the threshold for initiation of action potentials (Fig. 1, panels c, e, and legend). We have clarified this by amending the legend of Fig. 1 to indicate (page 25):

(c) Cumulative probability distributions of the amplitude of each SAC-evoked PSP recorded under the indicated conditions, from the indicated number of paired recordings.

8. Fig. 2b and d, the polar plots should be rotated 180 degrees so that the vector sums of the spiking responses point to the same preferred direction shown in Fig 2a and 2c.

With respect we disagree. The polar plots represent analysis of light-evoked action potential output, and are standardly presented in the orientation presented. We have previously clarified the direction of light movement in Figure 2, panel e.

References:

1. Lee, S., Kim, K. & Zhou, Z.J. Role of ACh-GABA cotransmission in detecting image motion and motion direction. *Neuron* **68**, 1159-1172 (2010).
2. Pei, Z., et al. Conditional knock-out of vesicular GABA transporter gene from starburst amacrine cells reveals the contributions of multiple synaptic mechanisms underlying direction selectivity in the retina. *J Neurosci* **35**, 13219-13232 (2015).
3. Wei, W., Hamby, A.M., Zhou, K. & Feller, M.B. Development of asymmetric inhibition underlying direction selectivity in the retina. *Nature* **469**, 402-406 (2011).
4. Yonehara, K., et al. Spatially asymmetric reorganization of inhibition establishes a motion-sensitive circuit. *Nature* **469**, 407-410 (2011).
5. Lipin, M.Y., Taylor, W.R. & Smith, R.G. Inhibitory input to the direction-selective ganglion cell is saturated at low contrast. *J Neurophysiol* **114**, 927-941 (2015).
6. Weng, S., Sun, W. & He, S. Identification of ON-OFF direction-selective ganglion cells in the mouse retina. *J Physiol* **562**, 915-923 (2005).
7. Ariel, M. & Daw, N.W. Pharmacological analysis of directionally sensitive rabbit retinal ganglion cells. *J Physiol* **324**, 161-185 (1982).
8. Kittila, C.A. & Massey, S.C. Pharmacology of directionally selective ganglion cells in the rabbit retina. *J Neurophysiol* **77**, 675-689 (1997).
9. Park, S.J., Kim, I.J., Looger, L.L., Demb, J.B. & Borghuis, B.G. Excitatory synaptic inputs to mouse on-off direction-selective retinal ganglion cells lack direction tuning. *J Neurosci* **34**, 3976-3981 (2014).
10. He, S. & Masland, R.H. Retinal direction selectivity after targeted laser ablation of starburst amacrine cells. *Nature* **389**, 378-382 (1997).
11. Chiao, C.C. & Masland, R.H. Starburst cells nondirectionally facilitate the responses of direction-selective retinal ganglion cells. *J Neurosci* **22**, 10509-10513 (2002).
12. Grzywacz, N.M., Amthor, F.R. & Merwine, D.K. Necessity of acetylcholine for retinal directionally selective responses to drifting gratings in rabbit. *J Physiol* **512**, 575-581 (1998).
13. Sethuramanujam, S., et al. A central role for mixed acetylcholine/GABA transmission in direction coding in the retina. *Neuron* **90**, 1243-1256 (2016).
14. Williams, S.R. & Mitchell, S.J. Direct measurement of somatic voltage clamp errors in central neurons. *Nat Neurosci* **11**, 790-798. (2008).
15. Poleg-Polsky, A. & Diamond, J.S. Imperfect space clamp permits electrotonic interactions between inhibitory and excitatory synaptic conductances, distorting voltage clamp recordings. *PLoS One* **6**, 0019463 (2011).
16. Ford, K.J., Felix, A.L. & Feller, M.B. Cellular mechanisms underlying spatiotemporal features of cholinergic retinal waves. *J Neurosci* **32**, 850-863 (2012).
17. Briggman, K.L., Helmstaedter, M. & Denk, W. Wiring specificity in the direction-selectivity circuit of the retina. *Nature* **471**, 183-188 (2011).

Reviewers' Comments:

Reviewer #1 (Remarks to the Author):

In general, I have no concerns about the data in this study. However, it is important that the results of this manuscript are placed in the correct context of current understanding of retinal direction selectivity.

1. I am not sure if the authors understand that the reason that they see an asymmetry of SAC impact in the so-called "null and preferred DSGC dendritic subfield" is because the presynaptic SACs they recorded from overlapped with the DSGC only with one side of the SAC dendrites. It is not because the overlap occurs on the one side or the other of DSGC dendritic field. For example, a SAC whose soma is very close to the DSGC soma may have its dendritic tree covering both the null and preferred subfields of the DSGC dendrites. But the inhibitory synapses only occur from the SAC dendrites pointing to the null direction of the DSGC, which is now in the preferred subfield (see the sketch below on the right). So the DSGC dendritic field is homogenous. They cannot be divided into null and preferred subfields.

The authors argued: "The Previous work detailing the impact of SAC-mediated synaptic input to ON-OFF DSGCs have differentiated the preferred and null-sides of the dendritic arbor of ON-OFF DSGCs using an approach similar to the one illustrated and described in our manuscript 1-4."

Please note: in the first three references on On-Off DSGCs (Lee et al, Pei et al, and Wei et al), "the null side" and "the preferred side" mentioned in all three papers refer to the location of SAC somas relative to the DSGC somas, they are not used to divide DSGC dendritic field into two sides. To reiterate, yes, the term "preferred and null side of DSGCs" are standard in the field. However, they refer to the location of presynaptic SAC somas, not to the DSGC dendrites.

For a better understanding of this concept, please see Figure 6 in the review article about the DS circuit by Vaney, Sivyer and Taylor at Nature Reviews Neuroscience, 2012. This diagram clearly demonstrate that the null/preferred asymmetry does not occur in the DSGC dendrites, it is from the SAC dendrites. Another helpful figure is Figure 4 from Briggman et al, Nature 2011.

Therefore, the authors need to rename these terms. For example, the last sentence in page 2 can be changed to "dendritic synapses from SACs on the null side of DSGCs." In page 5, last paragraph, and Figure 2: change "the preferred subfield" and "the null subfield" to "the preferred side" and "the null side".

2. I am not convinced by the argument that the latency measurements by Lee et al, Neuron 2010

is an artifact of voltage clamp. In fact, the On layer of On-Off DSGC dendrites can be clamped very well. This is well-supported by experiments (Lee et al, Fig. 1C, note the complete blockade of cholinergic currents by the nicotinic antagonist Hex) and by computational modeling in the paper the author mentioned (Poleg-Polsky & Diamond, 2011, Figure 6c). On the other hand, although current clamp recording circumvents the issue of imperfect voltage control, it may potentially contaminate the GABAergic and (especially) the cholinergic responses with other voltage-gated conductances. This is indeed why the latency of synaptic currents is almost exclusively measured in voltage clamp.

3. Evidence for paracrine ACh signaling in the literature

The authors listed the following references to argue for the presence of evidence for paracrine ACh signaling. Unfortunately, none of these papers provide any direct evidence besides mere postulations.

Briggman et al., 2011: while this is a landmark paper, it is a connectomic study on the SAC-DSGC contacts. There is no functional or anatomical characterization of cholinergic synapses.

Sethuramanujam et al., Neuron: this paper does not have any data addressing the paracrine vs synaptic ACh signaling. The paracrine signaling is only postulated in Discussion. The references listed in their discussion unfortunately also contain no data to support paracrine ACh signaling. Ariel and Daw, 1982 and Schmidt et al, 1987 are two studies examining the firing properties of ganglion cells in the presence of cholinergic antagonists. Note that Ford et al., 2012 clearly demonstrate the non-synaptic release of ACh during stage II retinal waves. However, at this stage, many synaptic connections in the retina are not established or are immature. Indeed, the non-synaptic ACh-dependent retinal waves disappear before the maturation of the retinal circuitry and the onset of the light response. Therefore, it can instead be used to argue against paracrine action of ACh in the mature retina.

In contrast, direct evidence exists to support synaptic transmission using ACh between SACs and DSGCs: 1. The nicotinic EPSCs are fast, with a latency similar to other monosynaptic currents; 2. The release of ACh is calcium dependent; 3. The cholinergic synapses between SACs and DSGCs are equipped with synaptic proteins including presynaptic vesicular ACh transporter and postsynaptic ionotropic receptors. These are the standard definitions of synaptic transmission, but not paracrine release.

Reviewer #2 (Remarks to the Author):

I agree with the authors' responses to the comments of Reviewer #1. In my opinion, the authors addressed all relevant concerns in the previous round of revisions. They now adjust their

manuscript in a few places to accommodate further non-essential requests by Reviewer #1 and appropriately stand their ground on more substantial issues. I recommend publication of this manuscript.

Reviewer #1

We thank the reviewer for indicating "In general, I have no concerns about the data in this study".

We are entirely in agreement with the Reviewer that "However, it is important that the results of this manuscript are placed in the correct context of current understanding of retinal direction selectivity",

1. I am not sure if the authors understand that the reason that they see an asymmetry of SAC impact in the so-call "null and preferred DSGC dendritic subfield" is because the presynaptic SACs they recorded from overlapped with the DSGC only with one side of the SAC dendrites. It is not because the overlap occurs on the one side or the other of DSGC dendritic field. For example, a SAC whose soma is very close to the DSGC soma may have its dendritic tree covering both the null and preferred subfields of the DSGC dendrites. But the inhibitory synapses only occur from the SAC dendrites pointing to the null direction of the DSGC, which is now in the preferred subfield (see the sketch below on the right). So the DSGC dendritic field is homogenous. They cannot be divided into null and preferred subfields.

The authors argued: "The Previous work detailing the impact of SAC-mediated synaptic input to ON-OFF DSGCs have differentiated the preferred and null-sides of the dendritic arbor of ON-OFF DSGCs using an approach similar to the one illustrated and described in our manuscript 1-4."

Please note: in the first three references on On-Off DSGCs (Lee et al, Pei et al, and Wei et al), "the null side" and "the preferred side" mentioned in all three papers refer to the location of SAC somas relative to the DSGC somas, they are not used to divide DSGC dendritic field into two sides. To reiterate, yes, the term "preferred and null side of DSGCs" are standard in the field. However, they refer to the location of presynaptic SAC somas, not to the DSGC dendrites.

For a better understanding of this concept, please see Figure 6 in the review article about the DS circuit by Vaney, Sivyer and Taylor at Nature Reviews Neuroscience, 2012. This diagram clearly demonstrate that the null/preferred asymmetry does not occur in the DSGC dendrites, it is from the SAC dendrites. Another helpful figure is Figure 4 from Briggman et al, Nature 2011.

Therefore, the authors need to rename these terms. For example, the last sentence in page 2 can be changed to "dendritic synapses from SACs on the null side of DSGCs." In page 5, last paragraph, and Figure 2: change "the preferred subfield" and "the null subfield" to "the preferred side" and "the null side".

We thank the reviewer for this comment. We have used the terms preferred side and null side throughout the revised manuscript.

2. I am not convinced by the argument that the latency measurements by Lee et al, Neuron 2010 is an artifact of voltage clamp. In fact, the On layer of On-Off DSGC dendrites can be clamped very well. This is well-supported by experiments (Lee et al, Fig. 1C, note the complete blockade of cholinergic currents by the nicotinic antagonist Hex) and by computational modeling in the paper the author mentioned (Poleg-Polsky & Diamond, 2011, Figure 6c). On the other hand, although current clamp recording circumvents the issue of imperfect voltage control, it may potentially contaminate the GABAergic and (especially) the cholinergic responses with other voltage-gated conductances. This is indeed why the latency of synaptic currents is almost exclusively measured in voltage clamp.

We thank the reviewer for this comment. With respect we do not agree that "the On layer of On-Off DSGC dendrites can be clamped very well", based on our experimental findings in other dendritic neurons and the simulation results of Poleg-Polsky & Diamond, 2011¹. We have however very clearly indicated in the revised manuscript that our latency analysis is distinct to the results obtained by Lee et al. 2010. The results section of the manuscript has been revised (Page 5) to indicate:

We note, however, that previous paired recordings have revealed similar onset times of SAC-mediated cholinergic and GABAergic postsynaptic currents in DSGCs, when temporally overlapping excitatory and inhibitory synaptic currents were separated under somatic voltage-clamp by reversal potential²

3. Evidence for paracrine ACh signaling in the literature. The authors listed the following references to argue for the presence of evidence for paracrine ACh signaling. Unfortunately, none of these papers provide any direct evidence besides mere postulations. Briggman et al., 2011: while this is a landmark paper, it is a connectomic study on the SAC-DSGC contacts. There is no functional or anatomical characterization of cholinergic synapses. Sethuramanujam et al., Neuron: this paper does not have any data addressing the paracrine vs synaptic ACh signaling. The paracrine signaling is only postulated in Discussion. The references listed in their discussion unfortunately also contain no data to support paracrine ACh signaling. Ariel and Daw, 1982 and Schmidt et al, 1987 are two studies examining the firing properties of ganglion cells in the presence of cholinergic antagonists. Note that Ford et al., 2012 clearly demonstrate the non-synaptic release of ACh during stage II retinal waves. However, at this stage, many synaptic connections in the retina are not established or are immature. Indeed, the non-synaptic ACh-dependent retinal waves disappear before the maturation of the retinal circuitry and the onset of the light response. Therefore, it can instead be used to argue against paracrine action of ACh in the mature retina. In contrast, direct evidence exists to support synaptic transmission using ACh between SACs and DSGCs: 1. The nicotinic EPSCs are fast, with a latency similar to other monosynaptic currents; 2. The release of ACh is calcium dependent; 3. The cholinergic synapses between SACs and DSGCs are equipped with synaptic proteins including presynaptic vesicular ACh transporter and postsynaptic ionotropic receptors. These are the standard definitions of synaptic transmission, but not paracrine release.

We thank the reviewer for their detailed comments about previous evidence concerning paracrine-like transmission between SACs and postsynaptic DSGCs. We agree that previous data is only suggestive of the role of paracrine-like neurotransmission. We note however that the points raised by the reviewer do not preclude our interpretation, specifically we provide positive evidence for vesicular ACh release in our work, and demonstrate the activation of ionotropic postsynaptic AChRs. We note that the calcium sensitivity of release does not suggest that paracrine-like transmission is not operational, in this regard previous work, for example Lee et al. 2010², has shown a different calcium sensitivity for ACh and GABA release from SACs, suggesting potentially a different mode of neurotransmission. We also note that previous work using optogenetic techniques has demonstrated the synaptic and paracrine-like release of ACh in the neocortex, which are both action potential dependent, and both activate postsynaptic nAChRs³. Indeed, Bennet et al. 2012 have demonstrated that manipulation of ACh hydrolysis allows the differentiation between synaptic and paracrine-like release, showing that blockade of endogenous AChE impacts only postsynaptic responses generated by paracrine-like transmission. Moreover, Bennet et al. 2012 demonstrate that the local delivery of exogenous AChE selectively depresses postsynaptic responses evoked by paracrine-like transmission³. We have therefore conducted new experiments (detailed on Page 7 and 8 of the results section of the revised manuscript, and illustrated in Sup. Fig. 6), to demonstrate that the local application of exogenous AChE at the same concentration as used by³ (0.4 U / μ l) reversibly decreased the amplitude of pharmacologically isolated nAChR-mediated PSPs in paired ON-SAC-ON-DSGC recordings. In contrast control application experiments were without affect. Furthermore we have demonstrated that blockade of endogenous AChE, with ambenonium (50 nM), enhanced both light responses and unitary SAC-evoked excitatory nAChR-mediated PSPs (Figure 4). We indicate in the revised manuscript, in line with previous work, that the bi-directional sensitivity of unitary ON-SAC evoked excitatory PSPs provides direct evidence in support of a paracrine-like mode of neurotransmission between SACs and

DSGCs. We have amended the Results (Page 7-8) and Discussion (Page 17) section of the revised manuscript to indicate these new findings:

Previous work has demonstrated that the pharmacological manipulation of AChE allows investigation of the spatial relationship between cholinergic release sites and postsynaptic nAChRs, finding that manipulation of ACh hydrolysis alters cholinergic signalling only when release sites are spatially distant to activated postsynaptic nAChRs^{3, 4}. We therefore examined if blockade of AChE controlled pharmacologically isolated unitary nAChR-mediated PSPs evoked in paired SAC-DSGC recordings. Under these conditions the application of the AChE inhibitor ambenonium significantly enhanced unitary cholinergic excitatory transmission, but did not affect pharmacologically isolated unitary GABAergic inhibition (**Fig. 4c,d**; excitatory PSP integral: control= 20.8 ± 4.2 μ V.s; ambenonium= 54.3 ± 11.5 μ V.s; $P= 0.011$, $T= 4.49$; $n= 5$). Furthermore paired SAC-DSGC recording revealed that the augmentation of ACh hydrolysis by the local application of exogenous AChE reversibly attenuated the amplitude of pharmacologically isolated nAChR-mediated PSPs (**Supplementary Fig. 6**; AChE (0.4 U per μ l) dissolved in Ames solution: control= 1.87 ± 0.25 mV; AChE puff = 1.10 ± 0.17 mV; $P= 0.0023$, $T= 5.72$; $n= 6$). In contrast, the control local application of Ames solution did not alter SAC-DSGC excitatory transmission (**Supplementary Fig. 6**; control= 2.15 ± 0.64 mV; Ames puff= 1.99 ± 0.67 mV; $P= 0.105$, $T= 2.09$; $n= 5$). The bi-directional control of unitary SAC-evoked cholinergic transmission by the augmentation and reduction of AChE activity is therefore consistent with a spatial separation between SAC release sites and activated postsynaptic AChRs^{3, 4}. To verify that ACh release controls preferred and null direction light responses we depleted presynaptic ACh by blocking the vesicular ACh transporter with vesamicol⁵. When presynaptic ACh release was depleted, preferred and null direction light responses were severely attenuated, and SAC-DSGC excitatory synaptic transmission selectively depressed (**Fig. 4e-h**; voltage integral of median filtered light responses: preferred direction: control= 14.5 ± 3.4 mV.s; vesamicol= -3.3 ± 1.5 mV.s; $P= 0.0008$, $T= 9.05$; null direction: control= $4.9 \pm$

3.2 mV.s; vesamicol= -11.1 ± 1.9 mV.s; $P= 0.0004$, $T= 11.11$; $n= 5$; excitatory PSP amplitude: control= 3.0 ± 0.6 mV; vesamicol= 0.9 ± 0.2 mV; $P= 0.004$, $T= 4.47$; $n= 7$). Taken together these data reveal that ACh release from ON-SACs powerfully controls the physiological responsiveness of ON-DSGCs through the activation of postsynaptic nAChRs, in a manner consistent with a local paracrine form of neurotransmission.

In addition we have amended the Discussion section (Page 17) to indicate:

As our paired recordings, and previous results, have revealed the obligatory co-release of ACh and GABA from SACs, these data suggest that ACh may be released from, or have postsynaptic impact at, sites other than dendro-dendritic synapses, a finding that is supported by the relatively long time to onset of cholinergic PSPs, and the effects of manipulating ACh hydrolysis^{3,4}.

References

1. Poleg-Polsky, A. & Diamond, J.S. Imperfect space clamp permits electrotonic interactions between inhibitory and excitatory synaptic conductances, distorting voltage clamp recordings. *PLoS One* **6**, 0019463 (2011).
2. Lee, S., Kim, K. & Zhou, Z.J. Role of ACh-GABA cotransmission in detecting image motion and motion direction. *Neuron* **68**, 1159-1172 (2010).
3. Bennett, C., Arroyo, S., Berns, D. & Hestrin, S. Mechanisms generating dual-component nicotinic EPSCs in cortical interneurons. *J Neurosci* **32**, 17287-17296 (2012).
4. Lamotte d'Incamps, B., Krejci, E. & Ascher, P. Mechanisms shaping the slow nicotinic synaptic current at the motoneuron-renshaw cell synapse. *J Neurosci* **32**, 8413-8423 (2012).
5. Zhou, F.M., Liang, Y. & Dani, J.A. Endogenous nicotinic cholinergic activity regulates dopamine release in the striatum. *Nat Neurosci* **4**, 1224-1229 (2001).